

# Massive superfluid vortices and vortex necklaces on a planar annulus

Matteo Caldara[1], Andrea Richaud[1,2*],
Massimo Capone[1,3] and Pietro Massignan[2]

**1** Scuola Internazionale Superiore di Studi Avanzati (SISSA),
Via Bonomea 265, I-34136, Trieste, Italy
**2** Departament de Física, Universitat Politècnica de Catalunya,
Campus Nord B4-B5, E-08034 Barcelona, Spain
**3** CNR-IOM Democritos, Via Bonomea 265, I-34136 Trieste, Italy

⋆ arichaud@sissa.it

## Abstract

We study a superfluid in a planar annulus hosting vortices with massive cores. An analytical point-vortex model shows that the massive vortices may perform radial oscillations on top of the usual uniform precession of their massless counterpart. Beyond a critical vortex mass, this oscillatory motion becomes unstable and the vortices are driven towards one of the edges. The analogy with the motion of a charged particle in a static electromagnetic field leads to the development of a *plasma orbit theory* that provides a description of the trajectories which remains accurate even beyond the regime of small radial oscillations. These results are confirmed by the numerical solution of coupled two-component Gross-Pitaevskii equations. The analysis is then extended to a necklace of vortices symmetrically arranged within the annulus.

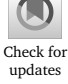

# 1  Introduction

The velocity flow in a superfluid is necessarily irrotational, implying that the circulation around a closed contour $\mathcal{C}$ is different from zero only if $\mathcal{C}$ encloses a phase singularity. Such a singularity generally corresponds to a vortex with a circulation which is an integer multiple of $h/m$ [1,2], being $h$ the Planck's constant and $m$ the mass of the particles in the superfluid.

In an unbounded superfluid in equilibrium, vortices form a two-dimensional rotating triangular lattice, which supports small-amplitude collective modes [3,4]: these vortex arrays have been observed in superfluid $^4$He [5] and also in cold atomic Bose-Einstein condensates (BECs) [6]. The determination of the equilibrium configuration in a superfluid with boundaries is more difficult, but it is related to the study of the dynamics of vortices in an incompressible nonviscous fluid. The latter traces back to the late 19$^{\text{th}}$ century with Ref. [7]: the motion of point vortices obeys *first* order equations where the $x$ and $y$ coordinates of each vortex serve as canonical variables. This description found wide applications first to superfluid $^4$He [8] and then also to dilute ultracold superfluid atomic BECs [9,10]. It is in this context that, besides the usual Hamiltonian formalism (see Sec. 157 of Ref. [7]), a powerful time-dependent Lagrangian approach was developed in Ref. [11] to study a ring of vortices in a BEC trapped in a circular container. At present, the study of the real-time dynamics of few-vortex systems is a very active field of research [12–14]. Additional states are possible in a multiply connected domain, consisting of the combination of vortices in the bulk of the fluid with circulation around the boundaries. Contrary to vortices in the fluid, the circulation along an inner boundary can have a quantum number much larger than one [15]. The dynamics of vortices on a planar annulus, one of the simplest realizations of a multiply connected domain, was first analyzed in Ref. [16] and then resumed in Ref. [17].

The first observation of a vortex in a cold dilute BEC took place at JILA with a bosonic mixture of two internal spin states of $^{87}$Rb. As explained in Ref. [18], coherent processes were used to create vortices in either of the two hyperfine components: one state supported a vortex, while the other (nonrotating) one played the role of a "defect" that filled the vortex core. Soon after, they also analyzed the dynamics and stability of vortices with a fraction of core particles varying from 10% to 50% [19]. This posed the question about the relevance of the effective mass of a vortex line and how it affects the dynamical properties of the vortex itself. Richaud *et al.* focused in Ref. [20] on vortices with massive cores in a two-dimensional

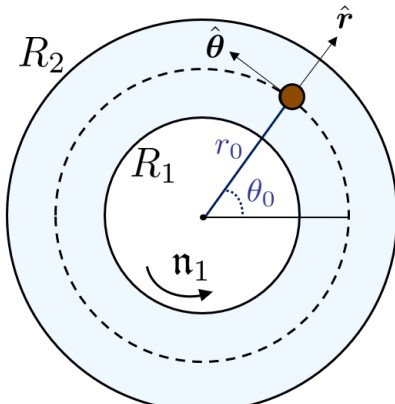

Figure 1: Schematic representation of the physical system for a single vortex ($N_v = 1$) inside a planar annulus with radii $R_1 < R_2$ and quantized flow circulation $\mathfrak{n}_1$ around the inner boundary. The superfluid $a$ component (light blue region) is confined inside the annulus and it contains a vortex with unit positive charge at position $\boldsymbol{r}_0 = (r_0, \theta_0)$. The $b$ component (brown circle) is trapped within the vortex core.

binary Bose mixture confined in a circular trap: the minority component trapped in the vortex cores provides an inertial mass that introduces *second* order acceleration terms, as in usual Newtonian mechanics. In the subsequent Ref. [21], the time-dependent variational Lagrangian method was used to derive the massive point-vortex model and to obtain various analytical predictions for the dynamics of two-component vortices with small massive cores.

The goal of this work is to analyze the dynamics of massive vortices on a planar superfluid film with annular geometry. This geometrical configuration is nowadays easily accessible using annular trapping potentials for ultracold atoms [23–28]. The physical system we focus on (see the sketch in Fig. 1) is a two-component mixture composed by $N_a \gg N_b$ particles with masses $m_a$ and $m_b$, which is confined in a planar annulus with inner and outer radii $R_1$ and $R_2$. The $a$ component is superfluid: it has a total mass $M_a = N_a m_a$, it contains $N_v$ identical vortices with unit positive charge and it features $\mathfrak{n}_1$ quanta of circulation around the inner radius $R_1$. Consequently, there are $\mathfrak{n}_1 + N_v$ quanta of circulation around the outer radius $R_2$. The $b$ component (which may or may not be superfluid) has total mass $M_b = N_b m_b$ and is trapped inside the vortex cores: this second species therefore provides each vortex with an effective core mass $M_c = M_b/N_v$. Fig. 1 shows the particular case of a single vortex ($N_v = 1$): the light blue area stands for the species $a$ that is spread inside the annular region hosting a vortex at position $\boldsymbol{r}_0$, while the brown circle denotes the species $b$ which is localized in the vortex core.

The organization of the work is the following. Section 2 contains the derivation of the massive point-vortex model from a variational Lagrangian approach and the analytical predictions for the dynamics of a single massive vortex. Earlier works [20, 21, 29] dealt with simply-connected geometries of the background superfluid, while here we focus on a ring geometry, which has a number of interesting features due to its non-trivial topology. Section 3 starts from the analogy with the familiar Lagrangian for a massive charged particle in a given electromagnetic field to develop a *plasma orbit theory*: this provides a framework to describe the trajectories of massive vortices beyond the regime of small radial oscillations. This theory represents the key novelty of this work and its validity can be extended to arbitrary planar geometries (included the disk one). The analytical predictions for one massive vortex are then compared in Section 4 with the numerical solution of the two-component Gross-Pitaevskii equation: the good agreement we find confirms that the point-vortex model provides an ac-

curate description of two-component vortices with small cores. Section 5 is devoted to the extension of our treatment to a necklace, *i.e.* a symmetric configuration of $N_v$ vortices in the annulus: the interest for this configuration is motivated by a strong experimental relevance, as it will be better explained in the following. Conclusions will be drawn in Section 6, together with an outlook on possible future extensions of the present work. The mathematical apparatus is rich of several technicalities which have been collected in the Appendices for sake of clarity. The material in the Appendices may safely be skipped, without compromising the physical understanding of the main concepts of this work. The dynamics of a single vortex without massive core was studied in Ref. [17] relying on the complex velocity potential: a generalization of this approach for a configuration of massless vortices in an annular geometry can be found in Appendix A. Appendices B and C review the main derivations for the model Lagrangian $\mathcal{L}_a$ and the *plasma orbit theory*, respectively. Appendix D, finally, develops the point-vortex model for a massive necklace.

## 2  Time-dependent variational Lagrangian method

Following the procedure outlined in Ref. [21], we use the time-dependent variational Lagrangian method (see Refs. [11, 30]) to obtain the Lagrangian of the system starting from simple trial quantum-mechanical wave functions.

For a one-component condensate wave function $\psi$, this method is based on a Lagrangian functional $\mathcal{L}$

$$\mathcal{L}[\psi] = \mathcal{T}[\psi] - \mathcal{E}[\psi], \tag{1}$$

where

$$\mathcal{T}[\psi] = \frac{i\hbar}{2} \int d^2r \left( \psi^*(\boldsymbol{r},t) \frac{\partial \psi(\boldsymbol{r},t)}{\partial t} - \frac{\partial \psi^*(\boldsymbol{r},t)}{\partial t} \psi(\boldsymbol{r},t) \right) \tag{2}$$

is the time-dependent part of the Lagrangian, the analog of kinetic energy in classical mechanics, and

$$\mathcal{E}[\psi] = \int d^2r \left( \frac{\hbar^2}{2m} |\nabla \psi(\boldsymbol{r},t)|^2 + V_{\text{tr}} |\psi(\boldsymbol{r},t)|^2 + \frac{g}{2} |\psi(\boldsymbol{r},t)|^4 \right) \tag{3}$$

is the customary Gross-Pitaevskii (GP) energy functional. It is easily proved that the Euler-Lagrange equation for the Lagrangian functional (1) corresponds to the time-dependent GP equation for $\psi$. Apart from being an exact approach, this Lagrangian formalism provides the basis for a powerful approximate variational method. If the trial wave function depends on several time-dependent parameters, the variational approach returns the dynamical equations governing their motion. For a system in equilibrium and stationary parameters, the resulting normal modes can be found from a next-order variation of the Lagrangian. In our case, the time-dependent parameters are the positions of the vortices $\{\boldsymbol{r}_j(t)\}_{j=1,\dots,N_v} \equiv \{\boldsymbol{r}_j(t)\}$.

### 2.1  Derivation of the massive point-vortex Lagrangian

The time-dependent variational Lagrangian $\mathcal{L}_a$ for the $a$ component can be derived using a trial wave function of the form

$$\psi_a(\boldsymbol{r}, \{\boldsymbol{r}_j\}) = \sqrt{n_a(\boldsymbol{r})} \, e^{i\mathcal{S}(\boldsymbol{r},\{\boldsymbol{r}_j\})}, \tag{4}$$

in terms of the density profile $n_a(\boldsymbol{r})$ and the phase $\mathcal{S}(\boldsymbol{r},\{\boldsymbol{r}_j\})$. The notation $(\boldsymbol{r},\{\boldsymbol{r}_j\})$ denotes a parametric dependence on the positions of all the vortices: they have the same charge $+1$ and polar coordinates on the plane are used in the following, *i.e.* $\boldsymbol{r}_j = (r_j, \theta_j)$, with $j = 1, 2, \dots, N_v$. This is precisely the approach followed in previous works with a rigid cylinder:

Refs. [21, 22] assume a uniform condensate, Refs. [11, 31] consider the more realistic Thomas-Fermi parabolic density profile, while Ref. [29] deals with generic $r^k$ potentials. For simplicity, we work here with a constant two-dimensional number density $n_a = N_a / [\pi(R_2^2 - R_1^2)]$. The method of images, well known from electrostatics [32], provides a convenient approach to satisfy the condition that the normal component of fluid velocity vanishes at all boundaries. While for a vortex inside a rigid cylinder a single image is sufficient, in an annulus an infinite series of images is needed since there are two boundaries: as specified in Ref. [16], there is actually a double infinite set of image vortices, beyond both the inner and outer edges of the annulus, that are arranged with alternating sign along the same radius as the physical vortex. In terms of polar coordinates on the plane $\boldsymbol{r} = (r, \theta)$, the resulting phase, as derived in Appendix A, reads:

$$\mathcal{S}(\boldsymbol{r}, \{\boldsymbol{r}_j(t)\}) = \mathfrak{n}_1 \theta + \sum_{j=1}^{N_v} \mathrm{Im} \left\{ \ln \left[ \frac{\vartheta_1 \left( \xi_j(\boldsymbol{r}), q \right)}{\vartheta_1 \left( \eta_j(\boldsymbol{r}), q \right)} \right] \right\}. \tag{5}$$

The first term accounts for the quantized flow circulation around the inner boundary of the annulus, while the second term encodes the contribution coming from the $N_v$ vortices and the corresponding images. The latter contains the Jacobi elliptic theta functions $\vartheta_1(z, q)$ that are integral functions of the complex variable $z$ and also depend on the geometric ratio:

$$q \equiv R_1 / R_2. \tag{6}$$

The arguments of the theta functions in Eq. (5) are defined in Eq. (A.6). The evaluation of $\mathcal{T}_a$ and $\mathcal{E}_a$ follows from Eqs. (2), (3) after inserting our trial wave function (4): the details of the derivation are presented in Appendix B, while here we show the final results. The first term is given by:

$$\mathcal{T}_a \left( \{ r_j, \dot{\theta}_j \} \right) = \pi \hbar n_a \sum_{j=1}^{N_v} \left( R_2^2 - r_j^2 \right) \dot{\theta}_j. \tag{7}$$

With our trial wave function, both the external potential energy and the mean field energy in Eq. (3) give a constant contribution that play an irrelevant role in the Lagrangian formalism. The quantity of interest is then the *energy difference* $\Delta \mathcal{E}_a$ between the vortex state and the vortex-free state. In our approximation, this difference corresponds to the kinetic energy of the physical vortices and their images integrated over the condensate density:

$$\Delta \mathcal{E}_a = \frac{\hbar^2 n_a}{2 m_a} \int_{ann} d^2 r \left| \boldsymbol{\nabla} \mathcal{S} \left( \boldsymbol{r}, \{ \boldsymbol{r}_j \} \right) \right|^2 = \int_{ann} d^2 r \, \frac{1}{2} m_a n_a \, \boldsymbol{v}^2, \tag{8}$$

where the subscript *ann* means that the integral is taken over the radial region $R_1 < r < R_2$. In evaluating the energy difference it is convenient to use a stream function $\chi$ together with the phase of the condensate wave function $\mathcal{S}$. It is also necessary to introduce a cut-off at the vortex core to ensure the convergence of the radial integral in Eq. (8). The lengthy analysis contained in Appendix B yields the compact result

$$\Delta \mathcal{E}_a \left( \{ \boldsymbol{r}_j \} \right) = \sum_{j=1}^{N_v} \Phi_j + \sum_{j,k=1}^{N_v} {}' V_{jk}, \tag{9}$$

where the primed sum means that we omit the terms $j = k$. The one-body term

$$\Phi_j = \Phi(r_j) \equiv \frac{\pi \hbar^2 n_a}{m_a} \left[ (1 - 2\mathfrak{n}_1) \ln \left( \frac{r_j}{R_2} \right) + \ln \left( \frac{2}{i} \frac{\vartheta_1 \left( -i \ln \left( \frac{r_j}{R_2} \right), q \right)}{\vartheta_1'(0, q)} \right) \right] \tag{10}$$

is the self-energy arising from the interaction of the vortex at $r_j$ with its infinite set of images, while the two-body term

$$V_{jk} = V(r_j, r_k) \equiv \frac{\pi \hbar^2 n_a}{m_a} \, \text{Re} \left[ \ln \left( \frac{\vartheta_1 \left( \eta_k(r_j), q \right)}{\vartheta_1 \left( \xi_k(r_j), q \right)} \right) \right] \tag{11}$$

is the interaction energy between vortices at $r_j$ and at $r_k$, including all their images (see Ref. [33] for the cylinder geometry). The $a$ component Lagrangian hence becomes:

$$\mathcal{L}_a = \sum_{j=1}^{N_v} \left\{ \pi \hbar n_a \left( R_2^2 - r_j^2 \right) \dot{\theta}_j - \Phi_j - \sum_{k=1}^{N_v} {}' V_{jk} \right\}. \tag{12}$$

For the species $b$ contribution $\mathcal{L}_b$, the trial wave function for the massive core is chosen to be a linear combination of Gaussian wave packets [30] localized at the positions of the vortices,

$$\psi_b(r, \{r_j\}) = \sum_{j=1}^{N_v} \left( \frac{N_b}{N_v \pi \sigma^2} \right)^{1/2} e^{-|r - r_j(t)|^2 / 2\sigma^2} e^{i r \cdot \alpha_j(t)}, \tag{13}$$

depending on $r_j(t)$ and $\alpha_j(t)$ as time-dependent parameters. The time-varying and space dependent overall phase ensures a non-zero superfluid velocity $\dot{r}_j(t) = \hbar \alpha_j(t)/m_b$, and the trial function is correctly normalized provided that the vortices are well-separated, *i.e.* for $|r_j - r_k| \gg \sigma$. A straightforward analysis (see Refs. [21, 30]) gives the corresponding Lagrangian:

$$\mathcal{L}_b = \sum_{j=1}^{N_v} \frac{M_b}{2N_v} \dot{r}_j^2. \tag{14}$$

Note that we always work in the *immiscible regime* where atoms of species $b$ only live inside the vortices of species $a$: this provides the physical justification to describe the species $b$-core with the same coordinates as the species $a$-vortex that hosts it.

It is useful to introduce dimensionless variables in terms of the properties of the $a$ component that contains the vortices, so that the resulting equations only depend on the mass ratio $\mu = M_b/M_a$. In particular, choosing the outer radius $R_2$ as the unit of length, $m_a R_2^2/\hbar$ as the unit of time and $\pi \hbar^2 n_a/m_a$ as the unit of energy, the model Lagrangian of the system $\mathcal{L} = \mathcal{L}_a + \mathcal{L}_b$ has the form:

$$\mathcal{L} = \sum_{j=1}^{N_v} \left[ \frac{\tilde{\mu}}{2N_v} \dot{r}_j^2 + \frac{r_j^2 - 1}{r_j^2} \, \dot{r}_j \times r_j \cdot \hat{z} - \Phi_j \right] - \sum_{j,k=1}^{N_v} {}' V_{jk}, \tag{15}$$

with $\tilde{\mu} = \mu(1 - q^2)$ and $\Phi_j$, $V_{jk}$ the dimensionless forms of Eqs. (10), (11).

The canonical angular momentum associated to the $j^{th}$ vortex is obtained from the model Lagrangian (15) as:

$$\ell_j = \frac{\partial \mathcal{L}}{\partial \dot{\theta}_j} = \frac{M_b}{N_v} r_j^2 \dot{\theta}_j + \pi \hbar n_a \left( R_2^2 - r_j^2 \right). \tag{16}$$

Here we restored physical units to make it clear that $\ell_j$ is made of the "mechanical" (Newtonian) contribution from species $b$ and the "vortex" contribution from species $a$: the latter, in particular, decreases as the vortex moves towards the outer boundary of the annulus. On the other hand, the angular momentum carried by each of the two species is directly computed

from the trial wave functions as

$$L_a = \langle \hat{L}_z \rangle_{\psi_a} = \pi \hbar n_a \left( R_2^2 - R_1^2 \right) \mathfrak{n}_1 + \pi \hbar n_a \sum_{j=1}^{N_v} \left( R_2^2 - r_j^2 \right), \tag{17}$$

$$L_b = \langle \hat{L}_z \rangle_{\psi_b} = \sum_{j=1}^{N_v} \frac{M_b}{N_v} r_j^2 \dot{\theta}_j, \tag{18}$$

where $\hat{L}_z = -i\hbar \, \partial / \partial \theta$ is the angular momentum operator. The total angular momentum of the system is then

$$L = L_a + L_b = \pi \hbar n_a \left( R_2^2 - R_1^2 \right) \mathfrak{n}_1 + \sum_{j=1}^{N_v} \ell_j, \tag{19}$$

where one recognizes a first term accounting for the quantized circulation around the inner ring and a second term due to the presence of $N_v$ quantized vortices inside the annulus. The symmetry of the Lagrangian (15) with respect to the polar angles of the vortices guarantees the total angular momentum to be a conserved quantity.

Before proceeding, the system under study admits three interesting simple limits that we briefly discuss considering a positive vortex at position $\boldsymbol{r}_j$ inside the annulus:

(i) when $R_1 \ll R_2$ and $\mathfrak{n}_1 = 0$, all the image vortices annihilate, except the one with negative charge at $\boldsymbol{r}_j{}' = (R_2/r_j)^2 \boldsymbol{r}_j$. The annulus reduces to a disk of radius $R_2$ where a single image vortex is required: the reader is referred to Appendix B of Ref. [17] for further details about the limit $R_1 \to 0$;

(ii) for $R_1 \to \infty$, keeping $R_2 - R_1 = D$ fixed, the curvature of the annulus becomes irrelevant and the system reduces to an infinitely long channel, or slab, with width $D$. As derived in detail in Refs. [34, 35], the vortex performs a uniform translation along the vertical direction with a different sign depending on the boundary it is closer to. This is a consequence of the equivalence between a vortex on a planar geometry and an electric charge, hence the interaction is a 2D Coulomb-like force scaling with the inverse of the distance. When the vortex is exactly in the middle of the slab, in particular, it doesn't move. Together with the infinite images, in fact, it forms a linear chain of equidistant alternating charges: the interactions from pair of symmetric images perfectly cancel each other and the dynamics is then inhibited;

(iii) when the vortex approaches either of the two boundaries, $r_j \simeq R_1$ or $r_j \simeq R_2$, the dominant contribution to the interaction comes from the image charge (with opposite sign) just beyond the closest edge, since all the others accumulate towards the centre of the annulus or far away from the outer border.

## 2.2 Dynamics of a single massive vortex

The starting point corresponds to the study of a single massive vortex inside the species $a$ at position $\boldsymbol{r} = (r, \theta)$. The Lagrangian (15) reduces to

$$\mathcal{L}(r, \dot{r}, \dot{\theta}) = \frac{1}{2} \tilde{\mu} \left( \dot{r}^2 + r^2 \dot{\theta}^2 \right) + \left( 1 - r^2 \right) \dot{\theta} - \Phi(r), \tag{20}$$

where we define the potential as

$$\Phi(r) \equiv (1 - 2\mathfrak{n}_1) \ln r + \ln \left( \frac{2}{i} \frac{\vartheta_1(-i \ln r, q)}{\vartheta_1'(0, q)} \right). \tag{21}$$

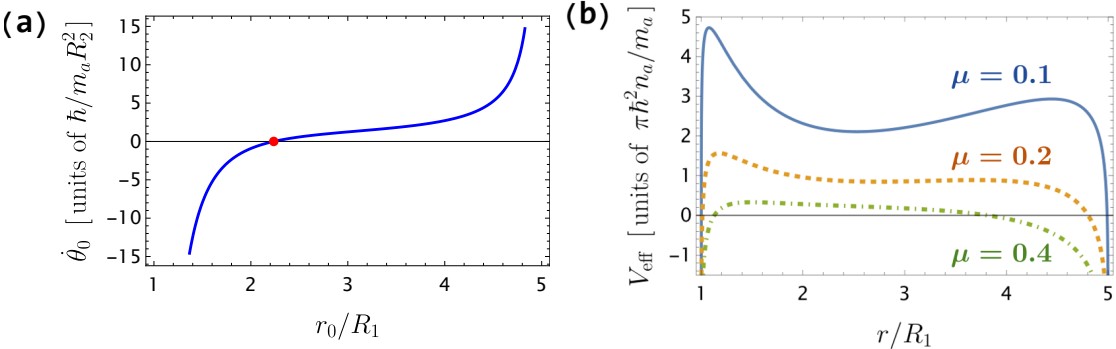

Figure 2: We consider an annulus with $R_2/R_1 = 5$ and $\mathfrak{n}_1 = 0$: these geometric parameters will not be changed in the rest of the work. (a) Uniform precession angular velocity for a massless vortex ($\mu = 0$). The red point marks the position where the angular velocity vanishes. (b) Effective potential for fixed $\ell = 0.75$ and increasing values of the mass ratio $\mu$. The smallest value $\mu = 0.1$ can support stable trajectories around the minimum $r_0$ of $V_{\text{eff}}$, while the curve with largest $\mu$ does not allow any stable trajectory.

Since the Lagrangian (20) does not depend on the polar angle $\theta$, the canonical angular momentum

$$\ell = \frac{\partial \mathcal{L}}{\partial \dot{\theta}} = \tilde{\mu} r^2 \dot{\theta} + 1 - r^2 \quad \Rightarrow \quad \ell = M_b r^2 \dot{\theta} + \pi \hbar n_a \left( R_2^2 - r^2 \right) \tag{22}$$

is a conserved quantity. On the right side of Eq. (22) we restored physical units to show that $\ell$ is a specific case of Eq. (16) for $N_v = 1$. The Euler-Lagrange equations

$$\tilde{\mu} \ddot{r} = \tilde{\mu} r \dot{\theta}^2 - 2 r \dot{\theta} - \Phi'(r), \qquad \tilde{\mu} r \ddot{\theta} = 2 \dot{r} \left( 1 - \tilde{\mu} \dot{\theta} \right) \tag{23}$$

are *second*-order differential equations in time. In the case of a massless vortex ($\mu = 0$) they reduce to *first*-order equations that determine the uniform precession of a massless vortex along an orbit of radius $r_0$ with constant angular velocity

$$\dot{\theta}_0 = -\frac{\Phi'(r_0)}{2 r_0} = \frac{\hbar}{m_a r_0^2} \left[ \mathfrak{n}_1 - \frac{1}{2} + \frac{i}{2} \frac{\vartheta_1'\left( -i \ln\left( \frac{r_0}{R_2} \right), q \right)}{\vartheta_1\left( -i \ln\left( \frac{r_0}{R_2} \right), q \right)} \right] . \tag{24}$$

The above result, given in conventional units, coincides with what was derived in Ref. [17] with a different approach based on the use of the complex velocity potential (here we briefly review it in Appendix A). Notice that the angular velocity (24) can change sign according to the radius of the circular orbit, as shown in Fig. 2(a). The physical intuition underlying this behaviour is related to the dominant role played by the closest image vortex with opposite sign. As the vortex approaches the outer boundary $R_2$, it mainly feels the interaction of the negative image charge beyond the edge: this situation is similar to the case of a circular trap, hence the rotation is counterclockwise ($\dot{\theta}_0 > 0$). Moving towards $R_1$, instead, the situation is reversed resulting into a clockwise rotation ($\dot{\theta}_0 < 0$). A similar discussion can be found in Appendix B of Ref. [17].

The conservation of angular momentum $d\ell/dt = 0$ allows to reduce Eqs. (23) to a single differential equation for $r(t)$ and develop the entire formalism based on the introduction of an effective potential $V_{\text{eff}}(r)$ [21, 29]. Following the same derivation as in Sec. III.A of Ref. [21], the explicit form of the effective potential turns out to be:

$$V_{\text{eff}}(r) = \frac{(\ell - 1)^2}{2 \tilde{\mu} r^2} + \frac{r^2}{2 \tilde{\mu}} + \Phi(r). \tag{25}$$

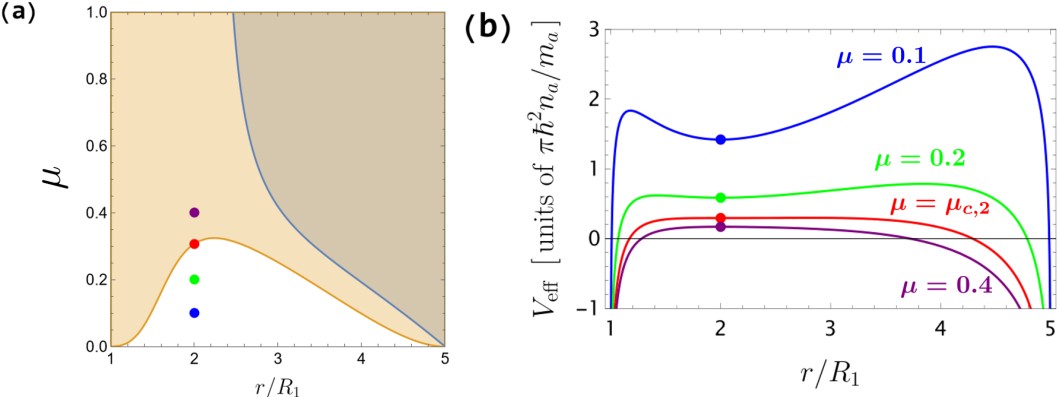

Figure 3: (a) Stability diagram of massive vortices. In the white region, massive vortices may perform small radial oscillations around the local minimum of $V_{\text{eff}}$. In the orange region ($\mu > \mu_{c,2}$), $V_{\text{eff}}$ only displays a local maximum, so that the vortex is rapidly expelled towards one of the two borders. In the light brown region on the right of the blue curve ($\mu > \mu_{c,1}$), the precession frequencies $\Omega_0^{(\pm)}$ become complex. (b) Effective potentials obtained for the four values of the mass ratio $\mu$ indicated by the corresponding dots in panel (a). For each curve, we chose a value of the angular momentum $\ell$ such that $V_{\text{eff}}$ has an extremum (minimum or maximum) at fixed $r/R_1 = 2$.

The first term is a repulsive centrifugal potential, while the last one is the analog of an attractive two-body central potential. The term in the middle, instead, plays the role of an attractive harmonic oscillator potential and it comes from the vortex contribution $(1 - r^2)\dot{\theta}$ in the Lagrangian (20). Fig. 2(b) shows typical plots of $V_{\text{eff}}$ for a fixed value of $\ell$ and different mass ratios $\mu$. For small $\mu$ and $\ell$ the effective potential for the annulus has one local minimum and two local maxima, since it diverges to $-\infty$ when approaching both the boundaries. As the mass ratio increases, the local minimum and the two maxima merge into a single maximum.

The motion at the local minimum corresponds to a uniform precession, but for massive cores there exist two solutions for the precession frequency:

$$\Omega_0^{(\pm)}(r_0) = \frac{1}{\tilde{\mu}}\left(1 \pm \sqrt{1 + \tilde{\mu}\frac{\Phi'(r_0)}{r_0}}\right). \tag{26}$$

When $\Phi'(r_0) > 0$ the two solutions are real for every $\mu$. Since $\Phi'(r_0) = -2r_0\dot{\theta}_0$, this happens whenever the precession frequency of the massless case $\dot{\theta}_0$ [which is plotted in Fig. 2(a)] is negative. In the small-mass limit the larger root $\Omega_0^{(+)} \approx 2/\tilde{\mu}$ diverges, becoming irrelevant, while the smaller root $\Omega_0^{(-)}$ reduces to the precession rate (24) for a massless vortex.

When $\dot{\theta}_0 > 0$ instead the roots become complex, signalling an instability, as soon as

$$\mu > \mu_{c,1} = -\frac{r_0}{(1 - q^2)\Phi'(r_0)} = \frac{1}{2(1 - q^2)\dot{\theta}_0}. \tag{27}$$

This unstable region is the light brown-shaded area in the phase diagram shown in Fig. 3(a).

A linear analysis of the perturbation around the precession radius $r_0$ yields the squared small-oscillations frequency

$$\omega^2 = \frac{4}{\tilde{\mu}^2}\left[1 + \frac{\tilde{\mu}}{4}\left(3\frac{\Phi'(r_0)}{r_0} + \Phi''(r_0)\right)\right]. \tag{28}$$

Notice that, for given $\ell$ and $\tilde{\mu}$, the energy is minimized when the system performs a simple uniform circular orbit: the onset of any oscillations corresponds to a dynamics with a higher total (conserved) energy. The small oscillations become unstable for mass ratios

$$\mu > \mu_{c,2} = -\left[\frac{1-q^2}{4}\left(3\frac{\Phi'(r_0)}{r_0} + \Phi''(r_0)\right)\right]^{-1}, \tag{29}$$

which defines the critical region represented by the orange-shaded area in Fig. 3(a). The diagram shows that Eq. (29) always provides a more restrictive condition than Eq. (27). In the white region in Fig. 3(a) both the uniform precession and small oscillations are allowed: consistently, the curves of the corresponding effective potential in Fig. 3(b) display a local minimum. In the region where the precession is stable but the small-oscillations are not, Fig. 3(b) shows that the local minimum turns into a local maximum, *i.e.* the orbit takes place on a classically unstable point: any arbitrary radial perturbation is enough to destabilize the uniform precession, leading to the expulsion of the vortex.

Notice also that the unique minimum of the effective potential gets deeper and deeper approaching the massless limit $\mu \to 0$: this explains why the small radial oscillations are not allowed in the massless case, where the velocity of a given vortex is simply determined by the superposition of the velocities generated by all the other (real and virtual) vortices.

In Fig. 4 we display representative trajectories for a single positive massive point vortex in a planar annulus with $q = 1/5$, using various mass ratios $\mu$. Fig. 4(a) shows the uniform precession on an orbit of radius $r_0 = 3R_1$. We then perturb the initial condition introducing a radial displacement $r_0 + \delta$. For small $\delta$ (compared to $r_0$), the corresponding trajectory, that is presented in Fig. 4(b), consists of small rapid stable oscillations superimposed on a slow precession. Fig. 4(c) is obtained for a larger displacement $\delta$ and the trajectory can be classified as an *epitrochoid*: this is a planar curve that is obtained from a smaller circle that rolls without sliding around the outside of a larger circle (a clear animation is found in Ref. [36]). This concept will be revised in the following section and we refer the reader to the last part of Appendix C for a more specific mathematical treatment. In general, an arbitrary initial radial displacement leads to such peculiar trajectories that cannot be regarded as small oscillations. We decide to denote them as *plasma orbits*, the reason being that they resemble the trajectory of a massive charged particle under the influence of an electromagnetic field: this analogy will be the central subject of Sec. 3. Finally, in Fig. 4(d) we show the dynamics when the mass of the species $b$ overcomes the critical value in Eq. (27): the vortex is expelled from the annulus following an unstable orbit.

## 3 Plasma orbit theory

The Lagrangian of a particle of mass $m$ and charge $Q$[1] in an electromagnetic field with scalar potential $\phi(\mathbf{r}, t)$ and vector potential $\mathbf{A}(\mathbf{r}, t)$ is given by

$$\mathcal{L}(\mathbf{r}, \dot{\mathbf{r}}, t) = \frac{1}{2}m\dot{\mathbf{r}}^2 + Q\,\dot{\mathbf{r}} \cdot \mathbf{A}(\mathbf{r}, t) - Q\,\phi(\mathbf{r}, t). \tag{30}$$

A direct comparison with the model Lagrangian (20) shows that a massive vortex with one quantum of circulation is equivalent to a particle with mass $m = \tilde{\mu}$ and charge $Q = +1$ moving inside a static electromagnetic field

$$\mathbf{E}(r) = -\boldsymbol{\nabla}\phi(\mathbf{r}) = -\frac{\pi\hbar^2 n_a}{m_a R_2}\Phi'(r)\,\hat{\mathbf{r}}, \qquad \mathbf{B} = \boldsymbol{\nabla} \times \mathbf{A}(\mathbf{r}) = -2\pi\hbar n_a\,\hat{\mathbf{z}}, \tag{31}$$

---

[1]We use $Q$ instead of the usual notation $q$ for the electric charge to avoid confusion with the geometric ratio introduced in Eq. (6).

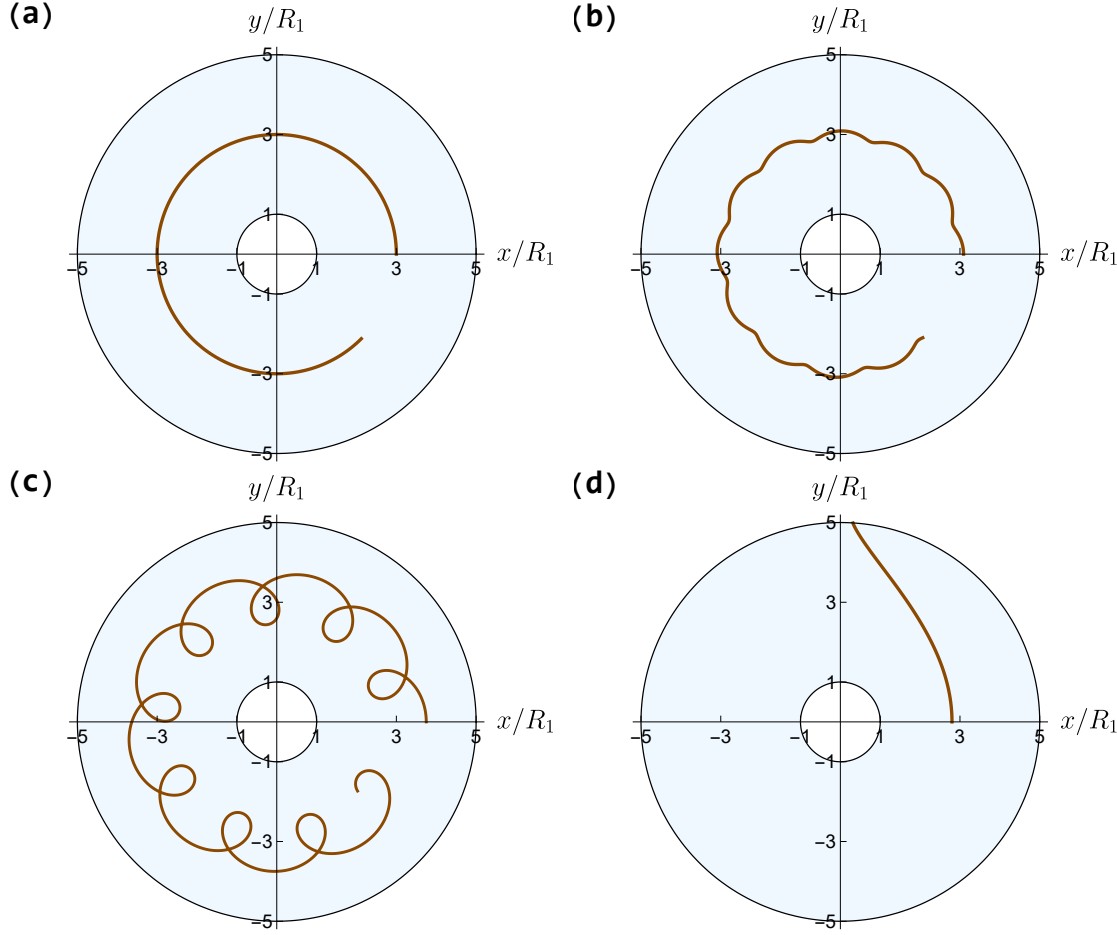

Figure 4: Four possible trajectories for a single positive massive point vortex confined in the same planar annulus as in Fig. 2. Curves correspond to the numerical solutions of the Euler-Lagrange equations (23). We consider a mass ratio $\mu = 0.1$. (a) Circular orbits followed with uniform angular velocity. (b) The presence of a core mass leads to small-amplitude radial oscillations for a little initial displacement from the precession orbit. (c) A *plasma orbit* appears for a larger initial radial displacement: the specific curve is an *epitrochoid* (further details in the main text). (d) For larger core mass, more precisely $\mu = 0.5 > \mu_{c,1}$, the massive vortex moves continuously towards the outer boundary where it is expelled.

where the second forms are expressed in conventional units.[2] Within this formal analogy, the massive vortex behaves as a massive charge which experiences an effective nonuniform electric field pointing in the radial direction and an effective uniform magnetic field normal to the plane and proportional to the density of species $a$ condensate inside the annulus. The charged particle is subject to the Lorentz force and it obeys the equations of motion, given in terms of the total velocity $\boldsymbol{v} = \dot{\boldsymbol{r}}$:

$$\tilde{\mu}\frac{d\boldsymbol{v}}{dt} = \boldsymbol{E}(r) + \boldsymbol{v} \times \boldsymbol{B}\,. \tag{32}$$

It is easy to prove that, after singling out the components in both the radial and tangential directions, one recovers the Euler-Lagrange eqs. (23) previously obtained within the Lagrangian formalism. Following a common approach in plasma physics, the overall motion of a charged

---

[2]$\Phi'(r)$ is here dimensionless.

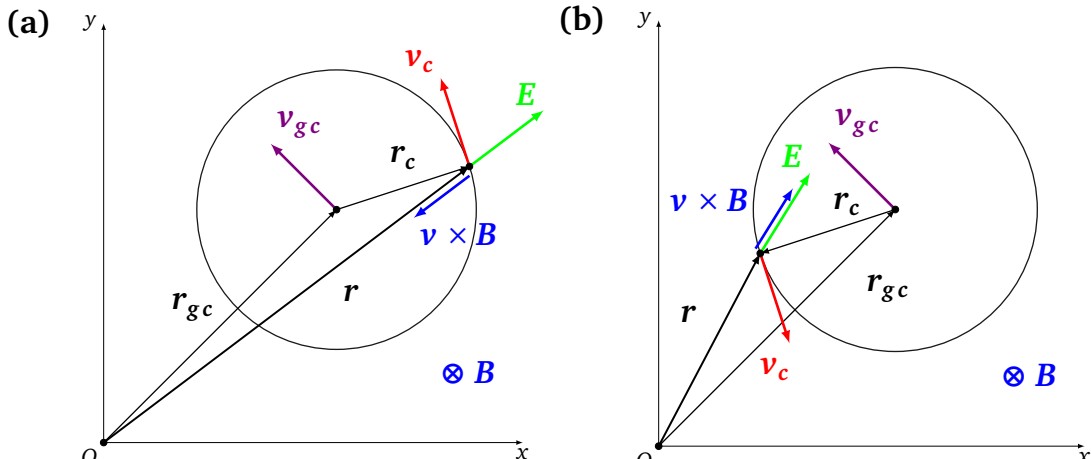

Figure 5: Schematic representation of the circular trajectory around the magnetic field lines followed by the massive vortex in the presence of uniform electric and magnetic fields. (a) On the first half of the orbit, the radial electric field opposes the magnetic force $v \times B$, resulting in a lower centripetal acceleration and a larger radius of curvature. (b) On the second half of the orbit the situation is the opposite: the electric and magnetic forces sum up to give a smaller radius of curvature.

particle inside an electromagnetic field can be thought as the combination of a *gyromotion*, *i.e.*, a circular motion with velocity $v_c$ around a central point called guiding centre (g.c.), and a translational motion of the guiding centre with velocity $v_{gc}$. Such a decomposition implies that the position of the particle can be written as

$$r = r_{gc} + r_c. \tag{33}$$

In the following we will present the main results. The guidelines for the derivations can be found in Appendix C, while we refer to Ref. [37] for a pedagogical treatment of the subject.

In the presence of a uniform electric and magnetic field, the decomposition $v = v_{gc} + v_c$ separates Eq. (32) into two independent equations for the two motions. The one for the *gyromotion*

$$\tilde{\mu} \frac{dv_c}{dt} = v_c \times B \tag{34}$$

describes a uniform circular orbit around the guiding centre characterized by the *cyclotron frequency* $\omega_c \equiv B/\tilde{\mu} = 2/\tilde{\mu}$ and the *Larmor radius* $r_L \equiv |v_c|/\omega_c$. Notice that the rotation of the charged particle is such that it generates a magnetic field that counteracts the external one: for the magnetic field in Eq. (31), a positive charge describes a counterclockwise rotation along the circular trajectory shown in Fig. 5. The second equation

$$\tilde{\mu} \frac{dv_{gc}}{dt} = E + v_{gc} \times B \tag{35}$$

describes the so called $E \times B$ *drift* of the guiding centre that moves with a uniform velocity $v_{gc} \propto E \times B$. The fields in Eq. (31) are such that the drift velocity is along the tangential direction: the guiding centre follows a precession orbit with radius $r_{gc}$ and uniform angular velocity $\Omega_{gc}$. The reason for this drift can be understood from the following physical picture. In the first half-cycle of the particle's orbit in Fig. 5(a), the electric force is opposite to the magnetic force $v \times B$: this causes a reduction of the total centripetal force that results in a larger $r_L$. In the second half-cycle in Fig. 5(b), instead, the electric and magnetic forces sum up to give a stronger centripetal acceleration that makes $r_L$ smaller. The difference in the

radius of curvature on the two sides of the orbit is responsible for the drift $v_{gc}$ and gives rise to peculiar *epitrochoidal* trajectories, as the one in Fig. 4(c).

The assumption of uniform electric field is satisfied when the particle follows exactly a circular orbit with radius $r_{gc}$: in such a case, there is no *gyromotion* and it is straightforward to verify that the precession frequency $\Omega_{gc}(r_{gc})$ is given by two solutions that coincide with the two uniform precession frequencies in Eq. (26), previously derived with the Lagrangian formalism. Whenever a particle deviates from the orbit of radius $r_{gc}$, it experiences a non-uniform electric field that changes in magnitude and direction according to its spatial position: the decomposition in Eq. (33) is not exact and both the *gyrofrequency* and the uniform precession frequency gain corrections with respect to the expressions in Eqs. (C.4), (C.9). Nonetheless, the superimposition of the *gyromotion* and the drift of the guiding centre still remains a valid approximation when the Larmor radius is much smaller than the typical length scale for the spatial variation of the electric field. Within this regime, one can apply the *undisturbed orbit approximation*[3] and expand the components of the electric field around the guiding centre position up to second order in the *gyroradius* $r_c \ll r_{gc}$: this procedure (see Appendix C for further details) introduces new corrective terms inside the equations of motion (32). An average of these equations over an entire cycle of the gyratory motion provides the following corrected form for the angular velocity of the uniform precession:

$$\Omega_{gc}(r_{gc}) = \frac{1}{\tilde{\mu}}\left(1 \pm \sqrt{1 + \tilde{\mu}\frac{\Phi'(r_{gc}) + \Delta_{gc}}{r_{gc}}}\right), \tag{36}$$

where the quantity $\Delta_{gc}$ is defined in Eq. (C.13) of Appendix C. Importantly, such a correction depends on the second derivative of the electric field and it reproduces the *finite Larmor radius effect* introduced in Ref. [37].

We test this prediction on the specific *plasma orbit* shown in Fig. 4(c). A numerical integration for an entire period gives a "numeric" precession frequency $\Omega_{\text{num}}/2\pi \approx 0.25378$ Hz. The maximum and minimum value of the radial coordinate give an estimate of the Larmor radius $r_L \approx 4.525\,\mu$m (see the final part of Appendix C for a more detailed explanation), from which we can extract the corrected frequency in Eq. (36) as $\Omega_{gc}/2\pi \approx 0.25384$ Hz. The latter result is indeed closer to $\Omega_{\text{num}}$ with respect to the bare prediction of the massive point-vortex model in Eq. (26), namely $\Omega_0^{(-)} \approx 0.23370$ Hz. As a second effect, the non-uniformity of the electric field is also responsible for a shift of the *gyrofrequency*, as shown in Ref. [38]. Focusing on the terms that average to zero over one cycle of the gyromotion, one can obtain an improved expression for the *gyrofrequency*:

$$\omega_c \approx \frac{2}{\tilde{\mu}}\sqrt{1 + \frac{\tilde{\mu}}{4}\left[3\frac{\Phi'(r_{gc})}{r_{gc}} + \Phi''(r_{gc})\right]} = \omega\,. \tag{37}$$

Within the *undisturbed orbit approximation*, the *gyromotion* frequency is corrected by the local value and the first derivative of the electric field, so to recover the frequency of small oscillations given in Eq. (28). We remark that, when deriving Eq. (28), the small oscillations were assumed to be straight and transverse, while the gyromotion is inherently circular.

As a final remark, the advantage in developing the *plasma orbit theory* is twofold. On the one hand, it yields a more realistic model since it accounts for the local curvature of the effective electric field that is missed by the point-vortex model, where only the local value $E(r_{gc})$ matters: this provides a quantitative correction to the frequency of uniform precession. On the other hand, it gives a qualitative explanation for the trajectories of a massive vortex that cannot be captured by the regime of small oscillations. As anticipated at the end of Sec. 2,

---

[3]This is the name used in Ref. [37], while it is defined as *small Larmor radius approximation* in Ref. [38].

the trajectories within the *plasma orbit* regime are *epitrochoids*: they arise from a peculiar combination of two circular motions that is compatible with the decomposition in Eq. (33) (see Appendix C for a more detailed description). This analogy was already realized in the literature in studies of the motion of charged particles along a plane orthogonal to a specific magnetic field configuration: a twisted magnetic flux tube is considered in Ref. [39], while Sec. II.A of Ref. [40] deals with a Penning trap. Moreover, the results in Eqs. (36) and (37) are valid for a general planar geometry, provided the correct identification of the one-body potential $\Phi(r)$. In particular, they can be applied to the circular trap discussed in Ref. [21].

## 4 Gross-Pitaevskii analysis

The massive point-vortex model developed in Sec. 2 requires the species $a$ to be superfluid, but it does not imply any condition on the species $b$ which provides the massive contribution to the vortices. Such a model can therefore be applied to a broad variety of systems. In this Section we focus on a heteronuclear dilute Bose mixture at zero temperature, where both species are superfluid and any normal fraction can be safely neglected. The interparticle interaction can be described at the mean field level by a contact pseudopotential (see Chapter 4 of Ref. [9]): the intra-species interaction constants $g_{a/b} = (4\pi\hbar^2/m_{a/b})a_{a/b}$ and the inter-species one $g_{ab} = (2\pi\hbar^2/m_{ab})a_{ab}$, with $m_{ab}$ the reduced mass, are proportional to the s-wave scattering lengths $a_{a/b}$, $a_{ab}$. The values of the three coupling constants can be experimentally tuned by means of Feshbach resonances [41]. Here we restrict to repulsive intra- and inter-species interactions that satisfy the *immiscible regime* condition $g_{ab} > \sqrt{g_a g_b}$, meaning that the two species are not spatially overlapping. As discussed in Sec. 12.1.1 of Ref. [42], this condition holds for a uniform system. For a trapped system (where the density distribution is not uniform), the condition is more complex, and it involves the number of atoms in the two components, as discussed for example in Ref. [43] (see also Ref. [44] for the case of a ring trimer geometry). However, since the confining potential is composed of hard walls and the majority component is in the Thomas-Fermi regime, our system is effectively uniform apart from those regions around the vortices. As such, the condition $g_{ab} > \sqrt{g_a g_b}$ is sufficient to ensure immiscibility.

Such a two-dimensional system is naturally embedded inside a three-dimensional world. A *quasi*-2D configuration can be achieved experimentally by applying a strong harmonic confining potential along the $z$ direction. The full 3D wave function of the gas then factorizes into a planar contribution and a narrow 1D Gaussian with a width $\sigma_z$ equal to the harmonic oscillator length along $z$. The $z$ degree of freedom is frozen and it can be integrated out leading to an effective 2D system described by planar coordinates $r = (x, y)$: this procedure, known as *dimensional reduction*, is outlined in Ref. [45] and it is explained in detail in Chapter III of Ref. [46]. Therefore, the Bose mixture admits as order parameters two 2D complex wave functions $\psi_a$, $\psi_b$, one for each species: they are related to the local number densities by $n_{a/b}(r) = |\psi_{a/b}(r)|^2$ so that they are normalized to the total number of particles. The dynamics of the system is thus governed by two coupled GP equations,

$$
\begin{aligned}
i\hbar\frac{\partial \psi_a(r,t)}{\partial t} &= \left[-\frac{\hbar^2\nabla^2}{2m_a} + V_\perp^a(r) + \frac{g_a}{d_z}|\psi_a(r,t)|^2 + \frac{g_{ab}}{d_z}|\psi_b(r,t)|^2\right]\psi_a(r,t), \\
i\hbar\frac{\partial \psi_b(r,t)}{\partial t} &= \left[-\frac{\hbar^2\nabla^2}{2m_b} + V_\perp^b(r) + \frac{g_{ab}}{d_z}|\psi_a(r,t)|^2 + \frac{g_b}{d_z}|\psi_b(r,t)|^2\right]\psi_b(r,t),
\end{aligned}
\tag{38}
$$

where $d_z = \sqrt{2\pi}\,\sigma_z$ is the effective thickness of the thin two-dimensional condensate, assuming an equal confining potential along $z$ for both species, while $V_\perp^{a/b}$ are the external confining potentials on the plane which imprint the annular potential.

**(a)**

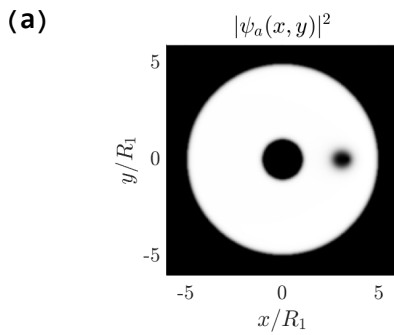

**(b)**

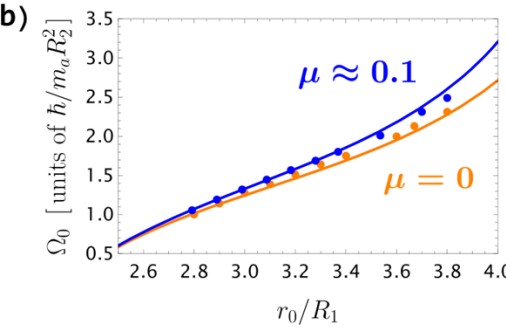

Figure 6: (a) Density of the $a$-component at the end of the imaginary-time evolution of the two coupled GP equations. Black (white) color corresponds to zero (high) values of the density. The model parameters employed for the simulation are reported in the text. (b) Comparison between the numerical results (points) and the analytical prediction (lines) for the precession frequency as a function of the radius of the orbit. For the massive case, $\Omega_0$ stands here for $\Omega_0^{(-)}$ in Eq. (26).

To test the predictions of the massive point-vortex model presented in Sec. 2.2, we performed numerical experiments with the two-component GP Eqs. (38). We consider an annulus with $R_1 = 10\,\mu\text{m}$, $R_2 = 50\,\mu\text{m}$, $d_z = 2\,\mu\text{m}$ and vanishing inner circulation $\mathfrak{n}_1 = 0$. The parameters for the two components are similar to the values in Refs. [20, 21]: $N_a = 5 \times 10^4$ $^{23}$Na atoms and $N_b \approx 1500$ $^{39}$K atoms, giving a mass ratio $\mu \approx 0.05$. The s-wave scattering lengths $a_a \approx 52\,a_0$, $a_b \approx 7.6\,a_0$, $a_{ab} \approx 24\,a_0$, where $a_0 \approx 5.29 \times 10^{-11}$ m is the Bohr radius, satisfy the *immiscibility* condition $g_{ab} > \sqrt{g_a g_b}$ [47]. We implement the simulation mapping the system on a $256 \times 256$ square grid with length $L_x = L_y = 120\,\mu\text{m}$, such that the grid spacing $\Delta x \simeq 0.5\,\mu\text{m}$ is smaller than the core size for a massless vortex, that can be estimated by the bare healing length $\xi = (8\pi n_a a_a)^{-1/2} \simeq 2\,\mu\text{m}$. The kinetic operators are implemented via FFT, while the time-dependent equations have been solved using a fourth order Runge-Kutta algorithm: the choice of the time step $\Delta t = 10\,\mu s$ is such that an excellent conservation of the total energy of the system is guaranteed during the time evolution. With this choice of the parameters one has that $\mu_a \ll \hbar\omega_z$ (being $\mu_a$ the chemical potential of species $a$), thus satisfying the assumption of a quasi-2D system, and $N_a a_a / d_z \gg 1$, corroborating the validity of the Thomas-Fermi approximation for the $a$ condensate. To generate the initial condition for our dynamics, we nucleate a vortex inside the species $a$ using a phase imprinting procedure. We also introduce a narrow and intense Gaussian pinning potential acting only on species $a$ and centred at the position of the vortex where a Gaussian peak in the species $b$ is also placed. We move to the frame rotating with angular velocity $\Omega$ adding the term $-\Omega \hat{L}_z$ to both Eqs. (38). The value of $\Omega$ is chosen accordingly to the radial position as given by the point-vortex model in Eq. (26). We perform an imaginary-time propagation in the rotating frame with this initial state, letting the system converge towards the ground state with the $b$-species core embedded in the $a$-species vortex: the ground state density of the $a$ component, that is shown in Fig. 6(a), is characterized by a vortex core which is broadened due to the presence of the $b$-atoms. Notice that the radius of the black circle represents an estimate for the core size of approximately $6\,\mu\text{m}$, corroborating the choice of the spatial mesh spacing. Subsequently, we turn off both the pinning potential and the rotation frequency to initiate a real-time propagation: at each time $t$, we track the position of the massive vortex by measuring the centre of mass of $|\psi_b|^2$.

As a first analysis, we study the uniform precession by fixing the radial position $r_0$ and measuring the corresponding angular velocity coming out from the real-time evolution. Fig. 6(b) shows how the points obtained from the numerical simulations nicely follow the analytical curves given by Eq. (24) for $N_b = 0$ (orange) and by Eq. (26) for $N_b \approx 3000$, *i.e.*, $\mu \approx 0.1$ (blue).

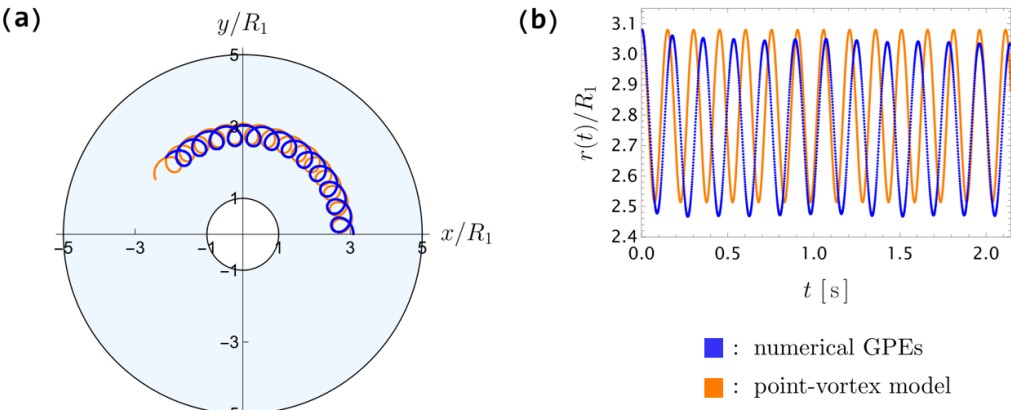

Figure 7: Numerical simulation of a *plasma orbit* obtained with the two-component GP numerical evolution (blue) compared with the analytical prediction of the massive point-vortex model (orange). The model parameters are the same as for Fig. 6, with $N_b \approx 1500$ ($\mu \approx 0.05$). (a) Trajectory of the vortex core. (b) Radial position of the vortex core.

Then, using the same numerical parameters introduced at the beginning of this section, we move to the analysis of *plasma orbits*. We start the imaginary time evolution by imprinting and pinning a massive vortex at $r_0^* = r_0 + \delta$ (with $|\delta| \ll r_0$) but we let the reference frame rotate at the precession frequency $\Omega_0^{(-)}(r_0)$ of the unshifted vortex. The subsequent propagation in real time shows that the vortex features radial oscillations superposed to the precession motion. In Fig. 7 we compare the numerical GP results (blue curve) with the prediction of the massive point-vortex model (orange curve) for a small $b$-species component $N_b \approx 1500$, corresponding to $\mu \approx 0.05$: the agreement is quite remarkable. This agreement is the natural consequence of having used the GP functional (3) in the time-dependent variational Lagrangian method explained in Sec. 2. The plot in Fig. 7(b) shows that the frequency of the radial oscillations in the numerical solution is slightly lower compared to the result of the point-vortex model. This discrepancy stems from the fact that the two-component GP equations describe two coupled many-body BECs with various internal modes that are completely missed by the point-vortex model, where the only degrees of freedom are the coordinates of the vortex core. The onset of vortex expulsion is also captured by numerical GP simulations: the agreement with the point-vortex model remains remarkable until the finite-sized vortex core touches either of the two boundaries.

# 5 Dynamics of a vortex necklace

In this section we consider the motion of a symmetric necklace composed by $N_v$ vortices that follow the same precession orbit with radius $r_0$ and uniform angular velocity $\Omega_{N_v}$. The polar coordinates of the $j^{th}$ vortex ( with $j = 1, 2, \ldots, N_v$) are given by

$$ r_j(t) \equiv r_0, \qquad \theta_j(t) = \varphi_0 + \frac{2\pi}{N_v}(j-1) + \Omega_{N_v} t, \qquad (39) $$

where $\varphi_0$ is an arbitrary initial phase for the first vortex that is irrelevant since only phase differences matter in Eq. (11).

For a massless necklace there is a unique angular velocity $\Omega_{N_v}(r_0)$ that can be derived with the complex potential formalism and it is given by Eq. (D.3) in Appendix D. It is shown

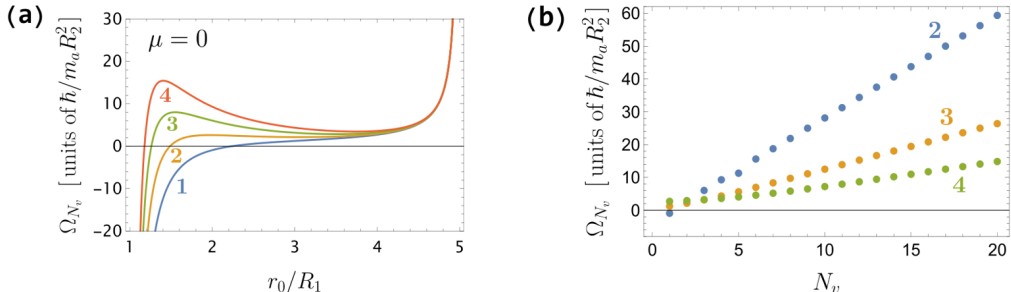

Figure 8: Uniform precession of a massless necklace for the same annular geometry as in Fig. 2. (a) Angular velocity as a function of the radius of the orbit for different numbers of vortices $N_v$ (the numbers label the value of $N_v$ for each curve). (b) Angular velocity as a function of the number of vortices for different radii of the orbit $r_0/R_1$ (the numbers refer to the value of $r_0/R_1$ for each curve).

in Fig. 8(a) as a function of the radius of the orbit $r_0$: each curve corresponds to a necklace with a different number of vortices $N_v$. All the curves are continuous, signalling that a uniform precession can take place at any radial position and with any number of vortices: depending on $r_0$, $\Omega_{N_v}$ can be positive, negative or even zero. The higher the number of vortices, the smaller is the region of clockwise precession. Fig. 8(b) is obtained taking vertical cuts of Fig. 8(a), *i.e.*, it shows the dependence of the angular velocity on the number of vortices at different fixed radii of the orbit $r_0/R_1$. For a fixed $N_v > 2$, the smaller the radius, the higher the velocity of the necklace; moreover, all the data sets display an almost linear behaviour for large $N_v$.

For a massive necklace, a careful analysis shows that the necklace can precess rigidly at frequencies (given here in conventional units)

$$\Omega_{N_v}^{(\pm)}(r_0) = \frac{\hbar}{m_a R_2^2} \frac{N_v}{\tilde{\mu}} \left(1 \pm \sqrt{1 - \frac{2\tilde{\mu} R_2^2}{N_v} \frac{\mathcal{B}(r_0)}{r_0^2}}\right). \tag{40}$$

See Appendix D.2 for the derivation of this result, and for the definition of $\mathcal{B}$. Exactly as in the case of a single massive vortex, there are two solutions for the precession frequency. In the following we consider only the physically relevant solution $\Omega_{N_v}^{(-)}$, which is well-behaved in the small-mass limit. In the presence of more than one vortex, the value of the mass ratio is chosen to scale linearly with $N_v$: in this way, the quantity $\mu/N_v$ fixes the amount of mass inside each vortex core for any arbitrary necklace. Now, the condition under which the uniform precession becomes unstable involves not only $\mu$ and $r_0$, but also $N_v$. To better understand the appearance of this instability, in Fig. 9(a) we fix the mass ratio to $\mu = 0.015 N_v$ and we look at the critical regions in the $(r_0, \mu)$ plane for various numbers of vortices. Next to it, Fig. 9(b) shows the dependence of the angular velocity on the radius of the orbit for different $N_v$. For a necklace of 6 vortices the red line corresponding to the fixed $\mu$ does not intersect the critical (green-shaded) area, but it is safely inside the stable region. This means that the uniform precession is allowed for every value of $r_0$ inside the annulus, in fact the green curve in Fig. 9(b) is a continuous curve qualitatively similar to the massless situation. This is true for all the necklaces with a small number of vortices, $1 \leq N_v \leq 6$, for this value of $\mu$. For a necklace of 7 or more vortices, instead, the horizontal red line crosses the critical (shaded) area: the two crossing points delimit the radial region within which the uniform precession is not allowed. Moving to the figure on the right, two vertical asymptotes develop in correspondence of these radial positions: for $N_v \geq 7$, $\Omega_{N_v}^{(-)}(r_0)$ becomes discontinuous and splits into two distinct branches.

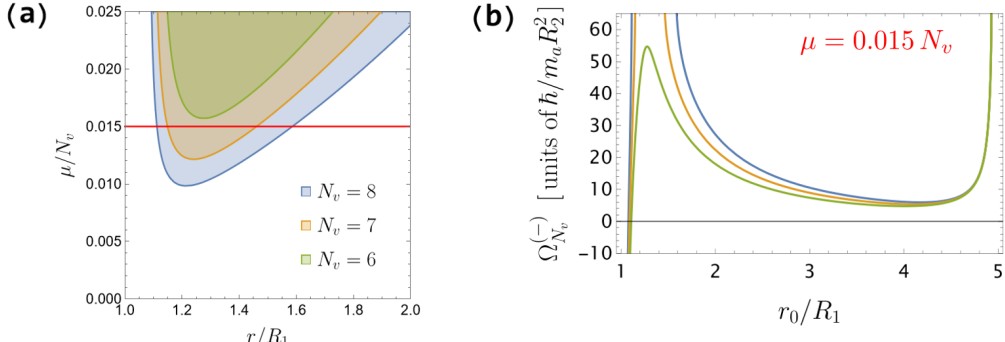

Figure 9: Uniform precession of a massive necklace for the same annular geometry as in Fig. 2. (a) Shaded areas represent regions of unstable uniform precession for different number of vortices $N_v$. The red horizontal line marks the selected value of the mass ratio $\mu = 0.015\,N_v$. (b) Angular velocity as a function of the radius of the orbit for different numbers of vortices $N_v$ (the choice of the colours is the same as in the left panel). An unstable region appears when $N_v \geq 7$.

As a benchmark for the previous discussion, we study the trajectories of a vortex necklace by numerically solving the equations of motion, as obtained from the model Lagrangian (15). We consider, in particular, a necklace made of $N_v = 4$ vortices, each of them carrying an amount of mass $\mu/N_v = 0.025$. Fig. 10(a) shows the uniform precession taking place along a circular orbit of radius $r_0 = 3R_1$ with constant angular velocity given by Eq. (40). We then modify the initial conditions, displacing radially outwards all vortices by a small quantity (from $r_0$ to $r_0 + \delta$), while maintaining fixed the angular momentum at the value which would give stable precession at the unshifted radial position $r_0$. The resulting trajectories in Fig. 10(b) are *epitrochoids*, like the one observed in Fig. 4(c) for the single vortex case. The necklace appears dynamically stable for a radially symmetric perturbation, however the study of the linear stability of the necklace and the possible chaotic regimes resulting from dynamical instabilities is at present an open and intriguing question.

# 6 Conclusions and outlook

We investigated the dynamics of vortices with empty and filled cores in a planar annulus geometry, motivated by the fact that the real-time dynamics of few-vortex systems is receiving considerable attention at present [12–14] and annular trapping potentials are within easy experimental reach [23–28]. We focused on the motion of vortices with massive cores and we obtained fully analytical predictions by means of a powerful point-vortex model. While a single massless vortex can only precess uniformly, the scenario is richer for a finite core mass. Uniform precession is allowed for small core masses, while it becomes unstable for large ones. The instability results in the collision of the massive vortex against either the inner or outer wall of the annular trap. This is explained in terms of an effective radial potential $V_{\text{eff}}(r)$ which depends on the mass ratio $\mu$ and the angular momentum $\ell$. Small radial oscillations are possible around the local minimum in $V_{\text{eff}}$, and we derived their frequency and stability with a linear theory in the perturbation.

Larger radial perturbations in the initial conditions lead to more peculiar trajectories, called *epitrochoids*, that constitute a non-trivial extension of the simpler small transverse radial oscillations. A new approach, borrowed from plasma physics (hence termed as *plasma orbit theory*), was developed to characterize them starting from the analogy with the Lagrangian of

**(a)**

**(b)**

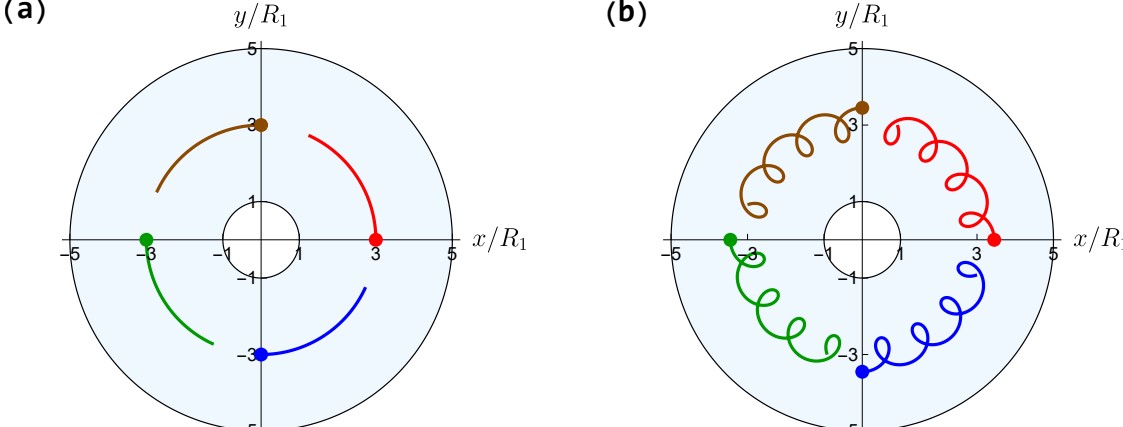

Figure 10: Trajectories for a necklace of $N_v = 4$ massive vortices obtained from a numerical solution of the equations of motion obtained from the Lagrangian (15). We consider a vortex mass $\mu/N_v = 0.025$. The dots stand for the initial position of each of the four vortices. (a) Circular orbits followed with uniform angular velocity. (b) When the dynamics is initiated with a radial position of the core which doesn't match the one needed for uniform precession (but maintaining the same angular momentum), massive vortices follow epitrochoidal trajectories.

a charged particle inside a transverse magnetic field and a radial, nonuniform electric field. For a weakly varying electric field, the *plasma orbit theory* provides results that recover the predictions from the point-vortex model, for the *gyromotion* frequency, and even improve them, for the precession frequency. Both these corrections result in a more refined model which, as long as the Larmor radius is small compared to the orbit radius, well captures the features of the vortex trajectories, both qualitatively and quantitatively.

We benchmarked the predictions of the massive point-vortex model against numerical simulations of the complete two-component GP equations. The analytical model finds a robust confirmation in the numerical results, as far as both the uniform precession and *plasma orbits* are concerned.

Finally, we generalized the analysis to a symmetric necklace of vortices. A neutral vortex necklace on a planar annulus was already studied in Ref. [48], while here all the vortices have unit positive charge. The presence of the mass leads to two possible roots for the precession frequency which become imaginary beyond a critical mass ratio. For a fixed mass ratio, instead, there exists a critical number of vortices which is connected to a region inside the annulus where stable precession orbits are not allowed (in contrast to a massless necklace). If the necklace is dynamically stable and the perturbation is radially symmetric, then each vortex of the necklace will have an epitrochoid-like orbit.

The present study suggests several interesting perspectives for future investigations. The planar annulus is topologically equivalent to a cylinder of finite length [17]: it is therefore very appealing to study the motion of massive vortices on cylindrical surfaces. In particular, it would be interesting to analyze the hydrodynamic analog of the Laughlin pumping [49] in such a geometry: as mentioned in Ref. [17], a slow pumping of angular momentum inside the system would allow a vortex to enter the lower rim of the cylinder, progressively spiral up and then leave it after reaching the upper rim, resulting in an increase by $\hbar$ of the total angular momentum per particle. Since the point-vortex model is very general, one can consider thin films of liquid helium with tracer particles instead of the binary BEC treated here in Sec. 4. Within this context, notice that Refs. [50–52] showed that the GP framework is able to reproduce many qualitative features of strongly interacting superfluid helium.

Another possible perspective is the characterization of the Kelvin-Helmholtz instability for a system of massless vortices. This instability is related to the elastic normal modes, known as Tkachenko modes [4,53], and it was recently investigated in a single-component atomic superfluid in Ref. [54]. We expect that the massive cores may alter profoundly the dynamics of this instability.

## Acknowledgements

The authors thank Alexander Fetter, Giacomo Roati and Francesco Scazza for enlightening discussions.

**Funding information** A. R. received funding from the European Union's Horizon research and innovation programme under the Marie Skłodowska-Curie grant agreement *Vortexons* no. 101062887. M. Cap. acknowledges support of MUR - Italian Minister of University and Research under the "Research projects of relevant national interest - PRIN 2020" - Project No. 2020JLZ52N, title "Light-matter interactions and the collective behavior of quantum 2D materials (q-LIMA)" and PRIN 2017 CEnTral (Protocol Number 20172H2SC4). M. Cap. further acknowledges financial support from MUR via PNRR MUR project PE0000023-NQSTI. P. M. and A. R. acknowledge support by grant PID2020-113565GB-C21 funded by MCIN/AEI/10.13039/501100011033, and by the Generalitat de Catalunya (Grant 2021 SGR 01411). P. M. further acknowledges support by the ICREA Academia program.

## A Complex potential theory for massless vortices on an annulus

An incompressible superfluid system can be described in terms of a macroscopic condensate wave function $\Psi = |\Psi| e^{i\mathcal{S}}$, whose phase $\mathcal{S}$ determines the superfluid velocity $\boldsymbol{v} = (\hbar/M)\boldsymbol{\nabla}\mathcal{S}$, being $M$ the atomic mass. The quantum-mechanical phase plays thus the role of the velocity potential and the irrotational condition is guaranteed (namely $\boldsymbol{\nabla} \times \boldsymbol{v} = 0$, except eventually at isolated points). Within the Thomas-Fermi regime, the local changes in the density of dilute ultracold superfluid BECs become small: the condition $\boldsymbol{\nabla} \cdot (n\boldsymbol{v}) = 0$, *i.e.* current conservation for a steady flow, then reduces to the incompressibility condition $\boldsymbol{\nabla} \cdot \boldsymbol{v} = 0$. An incompressible flow can be alternatively described by means of the stream function $\chi$. In particular, for a two-dimensional flow in the $xy$ plane, the velocity becomes

$$\boldsymbol{v} = (\hbar/M)\hat{\boldsymbol{n}} \times \boldsymbol{\nabla}\chi\,, \tag{A.1}$$

where $\hat{\boldsymbol{n}} = \hat{\boldsymbol{x}} \times \hat{\boldsymbol{y}}$ is the unit vector normal to the two-dimensional plane.

For an irrotational incompressible flow in two dimensions, the complex variable $z = x + iy$ provides a natural framework for the study of vortex dynamics. One can introduce a complex potential $F(z) = \chi(x, y) + i\mathcal{S}(x, y)$ which determines the hydrodynamic flow velocity components as:

$$v_y + iv_x = \frac{\hbar}{M}\frac{dF}{dz}\,. \tag{A.2}$$

A detailed discussion about the role of the complex potential can be found in Ref. [17]. In the same work, exploiting the fact that the surface of a cylinder of finite length is topologically equivalent to that of a planar annulus, the complex potential for a single positive vortex located

at $z_0 = x_0 + i y_0$ on a planar annulus of radii $R_1 < R_2$ is derived as:

$$F(z) = \mathfrak{n}_1 \ln\left(\frac{z}{R_2}\right) + \ln\left[\frac{\vartheta_1\left(-\frac{i}{2}\ln\left(\frac{z}{z_0}\right), q\right)}{\vartheta_1\left(-\frac{i}{2}\ln\left(\frac{z z_0^*}{R_2^2}\right), q\right)}\right]. \tag{A.3}$$

The Jacobi elliptic theta functions appear because an infinite set of image vortices is required to ensure that the normal component of the fluid velocity vanishes at the two boundaries (the inner and outer rings). The same complex potential had already been derived in Ref. [16] performing a conformal transformation on the solution for a line of equally spaced vortices between parallel boundaries. The two results in Refs. [16,17] are related by a Jacobi imaginary transformation (see Ref. [55] for more details on this) and differ by an overall factor $i$.

For a configuration of $N_v$ positive massless vortices inside a planar annulus at complex positions $z_j(t) = r_j(t) e^{i\theta_j(t)}$ $(j = 1, 2, \ldots, N_v)$, the complex potential comes from a straightforward generalization of Eq. (A.3):

$$F_{N_v}(z) = \mathfrak{n}_1 \ln\left(\frac{z}{R_2}\right) + \sum_{j=1}^{N_v} \ln\left[\frac{\vartheta_1\left(-\frac{i}{2}\ln\left(\frac{z}{z_j}\right), q\right)}{\vartheta_1\left(-\frac{i}{2}\ln\left(\frac{z z_j^*}{R_2^2}\right), q\right)}\right]. \tag{A.4}$$

The stream function for a system of vortices in an annulus is obtained as the real part of Eq. (A.4):

$$\chi(r, \theta) = \mathrm{Re}\, F_{N_v}(z)\bigg|_{z = r e^{i\theta}} = \mathfrak{n}_1 \ln\left(\frac{r}{R_2}\right) + \sum_{j=1}^{N_v} \mathrm{Re}\left[\ln\left(\frac{\vartheta_1\left(\xi_j(\boldsymbol{r}), q\right)}{\vartheta_1\left(\eta_j(\boldsymbol{r}), q\right)}\right)\right], \tag{A.5}$$

where we introduced

$$
\begin{aligned}
\xi_j(\boldsymbol{r}) &\equiv -\frac{i}{2}\ln\left(\frac{z}{z_j}\right)\bigg|_{z = r e^{i\theta}} = \frac{1}{2}\left(\theta - \theta_j(t)\right) - \frac{i}{2}\ln\left(\frac{r}{r_j(t)}\right), \\
\eta_j(\boldsymbol{r}) &\equiv -\frac{i}{2}\ln\left(\frac{z z_j^*}{R_2^2}\right)\bigg|_{z = r e^{i\theta}} = \frac{1}{2}\left(\theta - \theta_j(t)\right) - \frac{i}{2}\ln\left(\frac{r r_j(t)}{R_2^2}\right).
\end{aligned}
\tag{A.6}
$$

The imaginary part of the complex potential (A.4) gives the phase of the condensate wave function

$$\mathcal{S}(r, \theta) = \mathrm{Im}\, F_{N_v}(z)\bigg|_{z = r e^{i\theta}} = \mathfrak{n}_1 \theta + \sum_{j=1}^{N_v} \mathrm{Im}\left\{\ln\left[\frac{\vartheta_1\left(\xi_j(\boldsymbol{r}), q\right)}{\vartheta_1\left(\eta_j(\boldsymbol{r}), q\right)}\right]\right\}. \tag{A.7}$$

Using the results of Eqs. (A.5), (A.7) and exploiting some properties of theta functions, one can verify that the radial component of the flow velocity vanishes exactly at the borders of the annulus.

# B  Explicit calculation of the Lagrangian functional $\mathcal{L}_a$

In this Appendix we provide more details on the application of the time-dependent variational Lagrangian method discussed in Sec. 2. In particular, we focus on the derivation of the Lagrangian $\mathcal{L}_a$ for the species $a$ starting from the trial wave function (4), whose phase field is given in Eq. (5). The bulk of the mathematical calculations is based on what already done by Fetter in Ref. [16].

## B.1  Kinetic energy functional

Using the ansatz (4), the kinetic energy functional (2) becomes:

$$
\begin{aligned}
\mathcal{T}[\psi_a] &= -\hbar n_a \int_{ann} d^2 r \frac{\partial \mathcal{S}(\boldsymbol{r}, \{\boldsymbol{r}_j\})}{\partial t} \\
&= \frac{\hbar n_a}{2} \sum_{j=1}^{N_v} \mathrm{Im}\left[ \dot{\theta}_j \int_{R_1}^{R_2} dr\, r\, \mathcal{I}_j^{(-)}(r) - i\frac{\dot{r}_j}{r_j} \int_{R_1}^{R_2} dr\, r\, \mathcal{I}_j^{(+)}(r) \right],
\end{aligned}
\tag{B.1}
$$

where we introduced

$$
\mathcal{I}_j^{(\pm)}(r) \equiv \int_{-\pi}^{\pi} d\theta \left[ \frac{\vartheta_1'(\xi_j(\boldsymbol{r}), q)}{\vartheta_1(\xi_j(\boldsymbol{r}), q)} \pm \frac{\vartheta_1'(\eta_j(\boldsymbol{r}), q)}{\vartheta_1(\eta_j(\boldsymbol{r}), q)} \right].
\tag{B.2}
$$

Since the physical properties of the system are unchanged by a change of coordinate axis, the above integrals are independent of the angles $\theta_j$, which may be set to zero for convenience. We also set

$$
y_0 = -\ln(r/r_j)/2, \qquad y_1 = -\ln(rr_j/R_2^2)/2,
\tag{B.3}
$$

so that Eq. (B.2) becomes:

$$
\mathcal{I}_j^{(\pm)}(r) = 2 \int_{-\pi/2}^{\pi/2} dx \left[ \frac{\vartheta_1'(x + iy_0, q)}{\vartheta_1(x + iy_0, q)} \pm \frac{\vartheta_1'(x + iy_1, q)}{\vartheta_1(x + iy_1, q)} \right].
\tag{B.4}
$$

Let us start focusing on $\mathcal{I}_j^{(-)}(r)$. The integral is most simply evaluated by exploiting the periodicity of the theta functions in the complex plane. We shall therefore consider the following contour integral

$$
\oint_{\mathcal{C}} dz \frac{\vartheta_1'(z, q)}{\vartheta_1(z, q)},
\tag{B.5}
$$

taken over the rectangular path shown in Fig. 11(a), with corners at the points $z = \pm\pi/2 + iy_0$ and $z = \pm\pi/2 + iy_1$. The integrand function has simple poles that correspond to the simple zeros of $\vartheta_1(z, q)$: from Fig. 11(a) it is clear that the only relevant zero is at the origin, which lies inside the contour $\mathcal{C}$ only if $r > r_j$. The contour integral then reduces to

$$
\oint_{\mathcal{C}} dz \frac{\vartheta_1'(z, q)}{\vartheta_1(z, q)} = 2\pi i \operatorname*{Res}_{z=0}\left[ \frac{\vartheta_1'(z, q)}{\vartheta_1(z, q)} \right] \Theta(r - r_j) = 2\pi i\, \Theta(r - r_j),
\tag{B.6}
$$

where $\Theta$ is the Heaviside step function. The integral along the horizontal portions of the contour is just $\mathcal{I}_j^{(-)}(r)/2$, while the contribution from the vertical portions of the contour vanishes due to the periodicity of the theta function (see Ref. [55], p. 465). Substitutions inside Eq. (B.6) yield:

$$
\mathcal{I}_j^{(-)}(r) = 4\pi i\, \Theta(r - r_j).
\tag{B.7}
$$

As far as $\mathcal{I}_j^{(+)}(r)$ is concerned, by using the parity properties of the theta functions it can be rewritten as:

$$
\mathcal{I}_j^{(+)}(r) = 2 \int_{-\pi/2}^{\pi/2} dx \left[ \frac{\vartheta_1'(x + iy_0, q)}{\vartheta_1(x + iy_0, q)} - \frac{\vartheta_1'(x - iy_1, q)}{\vartheta_1(x - iy_1, q)} \right].
\tag{B.8}
$$

Considering now the rectangular path shown in Fig. 11(b), with corners at the points $z = \pm\pi/2 + iy_0$ and $z = \pm\pi/2 - iy_1$ and repeating the same procedure as before, one gets:

$$
\mathcal{I}_j^{(+)}(r) = -4\pi i\, \Theta(r_j - r).
\tag{B.9}
$$

After plugging Eqs. (B.7),(B.9) inside Eq. (B.1), the final expression for the kinetic energy functional as in Eq. (7) is obtained.

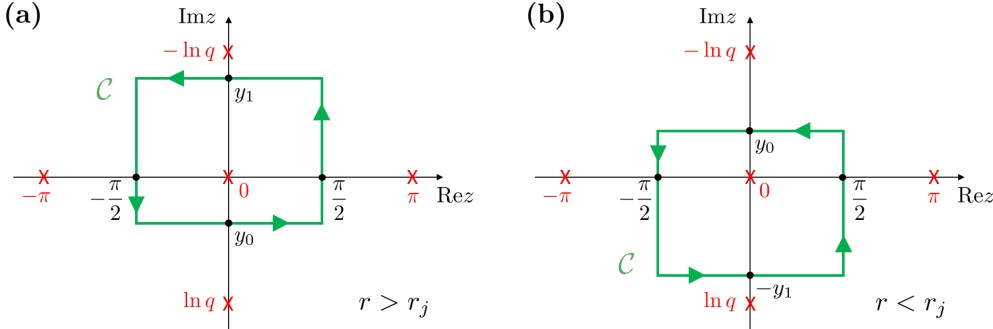

Figure 11: Integration contour $\mathcal{C}$ for the evaluation of the integrals in Eq. (B.2): red crosses represent the single poles of the integrand in Eq. (B.5), black points denote the coordinates of the corners of the rectangular path. (a) For $\mathcal{I}_j^{(-)}(r)$ the integration contour encloses the pole at $z = 0$ only if $r > r_j$. (b) For $\mathcal{I}_j^{(+)}(r)$ the contour contains the singularity at $z = 0$ only if $r < r_j$.

## B.2 Potential energy functional

The potential energy functional (8) is quadratic in the velocity field and it can be conveniently written in terms of the stream function

$$
\begin{aligned}
\Delta\mathcal{E}[\psi_a] &= \frac{\hbar n_a}{2}\int_{ann} d^2r \left(-v_x\frac{\partial\chi}{\partial y} + v_y\frac{\partial\chi}{\partial x}\right) \\
&= \frac{\hbar n_a}{2}\int_{ann} d^2r \left\{\left[\frac{\partial}{\partial x}(\chi\, v_y) - \frac{\partial}{\partial y}(\chi\, v_x)\right] - \chi\left(\frac{\partial v_y}{\partial x} - \frac{\partial v_x}{\partial y}\right)\right\} \\
&= \frac{\hbar n_a}{2}\left\{\oint_{\mathcal{C}_2} d\boldsymbol{l}\cdot\boldsymbol{v}\,\chi - \oint_{\mathcal{C}_1} d\boldsymbol{l}\cdot\boldsymbol{v}\,\chi - \int_{ann} d^2r\,\chi\,|\boldsymbol{\nabla}\times\boldsymbol{v}|\right\},
\end{aligned}
\tag{B.10}
$$

where the contours $\mathcal{C}_1$ and $\mathcal{C}_2$ are circles of radii $R_1$ and $R_2$, taken in the positive sense. The second line of Eq. (B.10) is obtained by partial integration, while the third line is an application of Green's theorem. Using Eq. (A.5), the line integral around $\mathcal{C}_2$ vanishes because $\chi(R_2, \theta) = 0$, while the line integral around $\mathcal{C}_1$ can be easily evaluated as:

$$
\begin{aligned}
\oint_{\mathcal{C}_1} d\boldsymbol{l}\cdot\boldsymbol{v}\,\chi &= R_1\,\chi(R_1)\int_{-\pi}^{\pi} d\theta\,v_\theta(R_1,\theta) \\
&= \frac{2\pi\hbar}{m_a}\left[\mathfrak{n}_1^2\ln\left(\frac{R_1}{R_2}\right) + \mathfrak{n}_1\sum_{j=1}^{N_v}\ln\left(\frac{r_j}{R_2}\right)\right].
\end{aligned}
\tag{B.11}
$$

The last term of Eq. (B.10) depends on the specific model adopted for the vortex core. We assume here that the vorticity $\boldsymbol{\zeta} = \boldsymbol{\nabla}\times\boldsymbol{v}$ is constant over a circular core of radius $a_c$ and vanishes outside of that region. The vortices do not overlap inside the annulus, therefore the integral simply reduces to a sum of terms evaluated at the position of each vortex

$$
\begin{aligned}
\int_{ann} d^2r\,\zeta\,\chi &= \sum_{j=1}^{N_v}\int_j d^2r\,\zeta\,\chi \\
&= \sum_{j=1}^{N_v}\int_j d^2r\,\zeta\,[\chi - \chi_{0j}] + \sum_{j=1}^{N_v}\int_j d^2r\,\zeta\,\chi_{0j},
\end{aligned}
\tag{B.12}
$$

where the subscript $j$ means that the integral is taken over the core of the $j^{th}$ vortex. The stream function displays a logarithmic divergence near the position of each vortex: in the second line of Eq. (B.12) the singular contribution of the $j^{th}$ vortex, denoted as $\chi_{0j}$, has been conveniently isolated. In the limit of vanishing core radius, the first term of Eq. (B.12) can be evaluated by setting $\zeta = 2\pi\hbar\delta(\boldsymbol{r} - \boldsymbol{r}_j)/m_a$, yielding

$$\sum_{j=1}^{N_v} \int_j d^2 r\, \zeta \left[\chi - \chi_{0j}\right] = \frac{2\pi\hbar}{m_a} \sum_{j=1}^{N_v} \chi_{(j)}, \tag{B.13}$$

where

$$\begin{aligned}
\chi_{(j)} &= \lim_{\boldsymbol{r}\to\boldsymbol{r}_j} \left[\chi(\boldsymbol{r}) - \ln\left(\frac{|\boldsymbol{r} - \boldsymbol{r}_j|}{a_c}\right)\right] \\
&= \mathfrak{n}_1 \ln\left(\frac{r_j}{R_2}\right) - \ln\left[\frac{2}{i}\frac{\vartheta_1\left(-i\ln\left(\frac{r_j}{R_2}\right), q\right)}{\vartheta_1'(0, q)}\frac{r_j}{a_c}\right] - \sum_{k(\neq j)=1}^{N_v} \mathrm{Re}\left[\ln\left(\frac{\vartheta_1(\eta_k(\boldsymbol{r}_j), q)}{\vartheta_1(\xi_k(\boldsymbol{r}_j), q)}\right)\right].
\end{aligned} \tag{B.14}$$

The last line of Eq. (B.14) is obtained using Eq. (A.5), while the explicit form of $\xi_k(\boldsymbol{r}_j)$ and $\eta_k(\boldsymbol{r}_j)$ comes straightforwardly from the definition in Eq. (A.6). The second term of Eq. (B.12) represents the small contribution to the energy from the core of each vortex. We develop the assumption of uniform vorticity by means of the *Rankine vortex model* that assumes a rigid body rotation inside a cylinder of radius $a_c$ and an irrotational flow outside of it. With such a model, the contribution from each vortex core is easily computed as:

$$\int_j d^2 r\, \zeta\, \chi_{0j} = -\frac{1}{4}\frac{2\pi\hbar}{m_a}. \tag{B.15}$$

A combination of Eqs. (B.10),(B.11),(B.14) and (B.15) leads to the final form for the potential energy functional:

$$\begin{aligned}
\Delta\mathcal{E}[\psi_a] = \frac{\pi\hbar^2 n_a}{m_a}\Bigg\{&\sum_{j=1}^{N_v}\left[-2\mathfrak{n}_1\ln\left(\frac{r_j}{R_2}\right) + \ln\left(\frac{2}{i}\frac{\vartheta_1\left(-i\ln\left(\frac{r_j}{R_2}\right), q\right)}{\vartheta_1'(0, q)}\frac{r_j}{a_c}\right) + \frac{1}{4}\right] \\
&+ \sum_{j,k=1}^{N_v}{}' \mathrm{Re}\left[\ln\left(\frac{\vartheta_1(\eta_k(\boldsymbol{r}_j), q)}{\vartheta_1(\xi_k(\boldsymbol{r}_j), q)}\right)\right] + \mathfrak{n}_1^2\ln\left(\frac{R_2}{R_1}\right)\Bigg\},
\end{aligned} \tag{B.16}$$

where the primed sum means that we omit the terms $j = k$. After simple algebraic manipulations and neglecting irrelevant constant contributions, one gets the final expression in Eq. (9):

$$\Delta\mathcal{E}_a\left(\{\boldsymbol{r}_j\}\right) = \sum_{j=1}^{N_v}\Phi_j + \sum_{j,k=1}^{N_v}{}' V_{jk}. \tag{B.17}$$

## C  *Plasma orbit theory*: detailed calculations

Restoring conventional units, the Lagrangian (20) for a single massive vortex reads:

$$\mathcal{L} = \frac{1}{2}M_b\dot{\boldsymbol{r}}^2 + \pi\hbar n_a\frac{r^2 - R_2^2}{r^2}\dot{\boldsymbol{r}}\times\boldsymbol{r}\cdot\hat{\boldsymbol{z}} - \Phi(r). \tag{C.1}$$

The comparison with Lagrangian (30) allows to identify the scalar and vector potentials

$$\phi(\boldsymbol{r}) = \Phi(r), \qquad \boldsymbol{A}(\boldsymbol{r}) = \pi\hbar n_a \frac{r^2 - R_2^2}{r^2} \boldsymbol{r} \times \hat{\boldsymbol{z}} = \pi\hbar n_a \left(\frac{R_2^2}{r} - r\right)\hat{\boldsymbol{\theta}}, \qquad \text{(C.2)}$$

from which the electric and magnetic fields in Eq. (31) are easily derived:

$$\boldsymbol{E}(r) = -\frac{\pi\hbar^2 n_a}{m_a R_2}\Phi'(r)\,\hat{\boldsymbol{r}}, \qquad \boldsymbol{B} = -2\pi\hbar n_a\,\hat{\boldsymbol{z}}. \qquad \text{(C.3)}$$

## C.1 Uniform electric field

Let us start with the analysis of the motion of a charged particle inside a uniform electric and magnetic field, following the pedagogical approach developed in Ref. [37]. On the one hand, Eq. (34) corresponds to a simple harmonic oscillator, with solution $\boldsymbol{v_c} = \boldsymbol{r_c} \times \boldsymbol{\omega_c}$, that describes a circular orbit around the guiding centre at the *cyclotron frequency*

$$\boldsymbol{\omega_c} \equiv \frac{\boldsymbol{B}}{\tilde{\mu}} = -\frac{2}{\tilde{\mu}}\,\hat{\boldsymbol{z}}. \qquad \text{(C.4)}$$

A further integration allows to get the coordinates of the *gyromotion*

$$x_c(t) = r_L \sin(\omega_c t), \qquad y_c(t) = r_L \cos(\omega_c t), \qquad \text{(C.5)}$$

where the *Larmor radius* quantifies the curvature of the trajectory and it is defined as

$$|\boldsymbol{r_c}| = r_L \equiv \frac{|\boldsymbol{v_c}|}{\omega_c}. \qquad \text{(C.6)}$$

On the other hand, Eq. (35) describes the $E \times B$ *drift* of the guiding centre that, due to the specific shape of the electromagnetic field in Eq. (31), undergoes a precession orbit with velocity

$$\boldsymbol{v_{gc}} = \boldsymbol{\Omega_{gc}} \times \boldsymbol{r_{gc}}, \qquad \text{(C.7)}$$

where $\boldsymbol{\Omega_{gc}} = \Omega_{gc}\hat{\boldsymbol{z}}$ is the constant angular velocity and $\boldsymbol{r_{gc}} = r_{gc}\hat{\boldsymbol{r}}$ is the guiding centre position.

When the particle follows exactly a circular orbit with radius $r_{gc}$, there is no *gyromotion* and it is possible to derive the expression of the precession frequency. The time derivative of Eq. (C.7) gives the centripetal acceleration of the guiding centre and, once plugged into Eq. (35), provides the following quadratic equation

$$\tilde{\mu}\Omega_{gc}^2 - B\Omega_{gc} + \frac{E(r_{gc})}{r_{gc}} = 0, \qquad \text{(C.8)}$$

whose two solutions coincide with the two uniform precession frequencies in Eq. (26):

$$\Omega_{gc}(r_{gc}) = \frac{1}{\tilde{\mu}}\left(1 \pm \sqrt{1 + \tilde{\mu}\frac{\Phi'(r_{gc})}{r_{gc}}}\right) \equiv \Omega_0^{(\pm)}(r_{gc}). \qquad \text{(C.9)}$$

## C.2 Corrections due to a non-uniform electric field

Whenever the particle deviates from the orbit of radius $r_{gc}$, then it experiences a non-uniform electric field that changes in magnitude and direction: in the following we will provide more

details about the derivations of these corrections within the *undisturbed orbit approximation* introduced in Sec. 3.

For each component $i = x, y$ of the electric field, the expansion up to second order in the *gyroradius* $r_c \ll r_{gc}$ reads:

$$E_i(r) = E_i(r_{gc}) + \nabla E_i\big|_{r_{gc}} \cdot r_c + \frac{1}{2} r_c \cdot \mathbf{H}[E_i]\big|_{r_{gc}} r_c + \mathcal{O}\left(r_c^3\right), \tag{C.10}$$

where $\mathbf{H}[E_i]$ is the Hessian matrix of the $i^{th}$ component and each spatial derivative is evaluated exactly at the position of the guiding centre. Notice that the electric field points in the radial direction and the above expansion is realized in cartesian coordinates: the zero-order terms then carry an angular part that is by itself dependent on the *gyroradius* components. As a first approximation, for the specific case of the $x$ component, one has:

$$E_x(r_{gc}) = E_x(r)\frac{x}{r}\bigg|_{r_{gc}} \simeq \frac{E_x(r_{gc})}{r_{gc}}\left(x_{gc} + x_c\right). \tag{C.11}$$

At this stage, the new equations of motion are obtained after plugging both the expansions (C.10), (C.11) inside Eq. (32).

The correction to the drift velocity can be evaluated by averaging the equations of motion over a cycle of the gyratory motion. Remembering the parametrization in Eq. (C.5), all the linear terms average to zero, while the only finite contributions are given by $\overline{x_c^2} = \overline{y_c^2} = r_L^2/2$, so that the drift velocity satisfies the equation

$$\tilde{\mu}\frac{d v_{gc}}{dt} = E(r_{gc})\frac{r_{gc}}{r_{gc}} + \frac{r_L^2}{4}\nabla^2 E\big|_{r_{gc}} + v_{gc} \times B. \tag{C.12}$$

A direct comparison with Eq. (35) shows that the correction depends on the second derivative of the electric field with a term that perfectly coincides with the *finite Larmor radius effect* introduced in Ref. [37]. The corrected form for the angular velocity of the uniform precession is then given in Eq. (36), where the quantity $\Delta_{gc}$ is defined as

$$-\frac{r_L^2}{4}\nabla^2 E\big|_{r_{gc}} = \frac{r_L^2}{4}\left[\Phi'''(r_{gc}) + \frac{1}{r_{gc}}\Phi''(r_{gc}) - \frac{\Phi'(r_{gc})}{r_{gc}^2}\right]\frac{r_{gc}}{r_{gc}} \equiv \Delta_{gc}\frac{r_{gc}}{r_{gc}}. \tag{C.13}$$

As a second effect, the non-uniformity of the electric field is also responsible for a shift of the *gyrofrequency*, as shown in [38]. Focusing on those terms that average to zero over one cycle of the gyromotion, one recovers the following linear equations of motion

$$\tilde{\mu}\ddot{r}_c = \begin{pmatrix} \partial_x E_x\big|_{r_{gc}} + \frac{E(r_{gc})}{r_{gc}} & \partial_y E_x\big|_{r_{gc}} \\ \partial_x E_y\big|_{r_{gc}} & \partial_y E_y\big|_{r_{gc}} + \frac{E(r_{gc})}{r_{gc}} \end{pmatrix} r_c + \dot{r}_c \times B, \tag{C.14}$$

that can be solved using the Laplace transform technique (see [38] for further details). The poles of the Laplace-transformed solution define the gyration of the particle around the guiding centre, providing the corrected expression for the *gyrofrequency* in Eq. (37):

$$\omega_c \approx \frac{2}{\tilde{\mu}}\sqrt{1 + \frac{\tilde{\mu}}{4}\left[3\frac{\Phi'(r_{gc})}{r_{gc}} + \Phi''(r_{gc})\right]} = \omega. \tag{C.15}$$

### C.3 Parametrization of the trajectories

The peculiar trajectories obtained in the *plasma orbit theory* regime, such as Fig. 4(c), belong to a particular class of plane curves, known as *epitrochoids*. Referring to the clear animation in Ref. [36], an *epitrochoid* is a *roulette* [56] traced by a point attached to a circle of radius $b$ rolling around the outside of a fixed circle of radius $a$, where the point is at a distance $h$ from the center of the exterior circle. The parametric equations are

$$
\begin{aligned}
x(\varphi) &= (a+b)\cos(\varphi) + h\cos\left(\frac{a+b}{b}\varphi\right), \\
y(\varphi) &= (a+b)\sin(\varphi) + h\sin\left(\frac{a+b}{b}\varphi\right),
\end{aligned}
\tag{C.16}
$$

and they resemble at once the superposition of two oscillatory motions. In particular, choosing $a+b = r_{gc}$, $h = r_c$ and $\varphi = \Omega_{gc}t$, one recovers the decomposition of the overall motion that is explained in Sec. 3:

$$
\begin{aligned}
x(t) &= r_{gc}\cos(\Omega_{gc}t) + r_c\cos(\omega_c t) = x_{gc}(t) + x_c(t), \\
y(t) &= r_{gc}\sin(\Omega_{gc}t) + r_c\sin(\omega_c t) = y_{gc}(t) + y_c(t).
\end{aligned}
\tag{C.17}
$$

The frequencies of the *gyromotion* and of the guiding centre are related by $\omega_c = r_{gc}\Omega_{gc}/b > \Omega_{gc}$: this is a consequence of the *pure rolling* of the circle of radius $b$ on the circle of radius $a$. The mathematical parameters $a$, $b$ and $h$ are physically related to the shape of the effective potential (that, in its turn, depends on the angular momentum $\ell$ and the mass ratio $\mu$) and the initial displacement $\delta$: however, there is not any easy way to establish a direct correspondence between them. Moreover, we stress again that the above decomposition is exact strictly in the presence of uniform fields: the nonhomogeneities of the electric field allow us only to provide approximate results. Nonetheless, Eq. (C.17) is the best proof to identify the type of trajectories beyond the *small oscillations* regime. Some trajectories correspond to an *epicycloid* [57], but it is simply a particular case of an *epitrochoid* with $b = h$. As a final comment, the identification of the trajectory allows to obtain an accurate estimate of the Larmor radius, that reduces to $r_L = h$. Eq. (C.17) imply that the radial coordinate oscillates in time between a maximum value $r_{\max} = r_{gc} + r_L$ and a minimum value $r_{\min} = r_{gc} - r_L$. Both $r_{\max}$ and $r_{\min}$ can be easily extracted from the numerical solution of the equations of motion, hence obtaining $r_L = (r_{\max} - r_{\min})/2$.

## D  Lagrangian approach for a necklace of massive vortices

In this Appendix we present the derivation of the angular velocity for the uniform precession of a vortex necklace on a planar annulus. We first present results for the massless case obtained with the complex potential formalism, and then we study the case of massive vortices by means of the Lagrangian approach.

### D.1  Massless necklace

The dynamics of $N_v$ massless vortices on a planar annulus can be studied by means of the complex potential formalism that we introduced in Appendix A. In particular, combining together Eqs. (A.2), (A.4) and developing some algebra, the complex velocity of the $k^{\text{th}}$ vortex

is obtained as

$$\dot{y}_k + i\dot{x}_k = \frac{\hbar}{m_a} \lim_{z \to z_k} \left[ \frac{dF_{N_v}(z)}{dz} - \frac{1}{z - z_k} \right]$$

$$= \frac{\hbar e^{-i\theta_k}}{m_a r_k} \left[ \mathfrak{n}_1 - \frac{1}{2} + \frac{i}{2} \frac{\vartheta'_1\left(-i\ln\left(\frac{r_k}{R_2}\right), q\right)}{\vartheta_1\left(-i\ln\left(\frac{r_k}{R_2}\right), q\right)} - \frac{i}{2} \sum_{j=1}^{N_v} {}' \left( \frac{\vartheta'_1\left(\xi_k(\mathbf{r}_j), q\right)}{\vartheta_1\left(\xi_k(\mathbf{r}_j), q\right)} - \frac{\vartheta'_1\left(\eta_k(\mathbf{r}_j), q\right)}{\vartheta_1\left(\eta_k(\mathbf{r}_j), q\right)} \right) \right]. \quad \text{(D.1)}$$

A simple case is given by a symmetric configuration of $N_v$ vortices undergoing a uniform precession with radius $r_0$ and angular velocity $\Omega_{N_v}$. All the vortices have the same velocity, therefore it is possible to write the complex positions as:

$$z_j(t) = r_0 e^{i\left(\Omega_{N_v} t + 2\pi(j-1)/N_v\right)}, \qquad j = 1, 2, \ldots, N_v. \quad \text{(D.2)}$$

Plugging this *ansatz* inside Eq. (D.1) and exploiting several properties of the Jacobi theta functions, one ends up with the following expression for the precession angular velocity of a necklace of $N_v$ massless vortices with radius $r_0$,

$$\Omega^0_{N_v}(r_0) = \frac{\hbar}{m_a r_0^2} \left[ \mathfrak{n}_1 - \frac{1}{2} + \frac{i}{2} \frac{\vartheta'_1\left(-i\ln\left(\frac{r_0}{R_2}\right), q\right)}{\vartheta_1\left(-i\ln\left(\frac{r_0}{R_2}\right), q\right)} + \frac{i}{2} \sum_{j=2}^{N_v} \left( \frac{\vartheta'_1\left(\alpha_j - i\ln\left(\frac{r_0}{R_2}\right), q\right)}{\vartheta_1\left(\alpha_j - i\ln\left(\frac{r_0}{R_2}\right), q\right)} \right) \right], \quad \text{(D.3)}$$

where we defined

$$\alpha_j = \frac{\pi}{N_v}(1-j). \quad \text{(D.4)}$$

In the presence of a single massless vortex, the summation gives no contribution and one recovers the expected result in Eq. (24):

$$\Omega^0_{N_v=1}(r_0) = \frac{\hbar}{m_a r_0^2} \left[ \mathfrak{n}_1 - \frac{1}{2} + \frac{i}{2} \frac{\vartheta'_1\left(-i\ln\left(\frac{r_0}{R_2}\right), q\right)}{\vartheta_1\left(-i\ln\left(\frac{r_0}{R_2}\right), q\right)} \right]. \quad \text{(D.5)}$$

## D.2  Massive necklace

The introduction of the mass requires to rely on the time-dependent variational Lagrangian method in order to study the corresponding dynamics. The starting point is the model dimensionless Lagrangian for a necklace of $N_v$ vortices that is explicitly written starting from Eq. (15) as:

$$\mathcal{L}_{N_v} = \sum_{j=1}^{N_v} \left\{ \frac{\tilde{\mu}}{2N_v} \left( \dot{r}_j^2 + r_j^2 \dot{\theta}_j^2 \right) + \left(1 - r_j^2\right) \dot{\theta}_j - \Phi(r_j) - \sum_{k=1}^{N_v} {}' \mathrm{Re} \left[ \ln\left( \frac{\vartheta_1\left(\eta_k(\mathbf{r}_j), q\right)}{\vartheta_1\left(\xi_k(\mathbf{r}_j), q\right)} \right) \right] \right\}. \quad \text{(D.6)}$$

We focus on the uniform precession of the symmetric configuration described in Sec. 5, therefore we make use of the *ansatz* (39). Since all the vortices have the same radial position and angular velocity, we may consider simply the Euler-Lagrange equations for the first one:

$$\frac{\tilde{\mu}}{N_v} \Omega^2_{N_v} - 2\Omega_{N_v} + \frac{2}{r_0^2} \left\{ \mathfrak{n}_1 - \frac{1}{2} + \frac{i}{2} \frac{\vartheta'_1(-i\ln r_0, q)}{\vartheta_1(-i\ln r_0, q)} \right.$$

$$\left. + \frac{1}{2} \sum_{j=2}^{N_v} \mathrm{Im} \left[ \frac{\vartheta'_1\left(\alpha_j, q\right)}{\vartheta_1\left(\alpha_j, q\right)} - \frac{\vartheta'_1\left(\alpha_j - i\ln r_0, q\right)}{\vartheta_1\left(\alpha_j - i\ln r_0, q\right)} \right] \right\} = 0, \quad \text{(D.7)}$$

$$\sum_{j=2}^{N_v} \mathrm{Re} \left[ \frac{\vartheta'_1\left(\alpha_j, q\right)}{\vartheta_1\left(\alpha_j, q\right)} - \frac{\vartheta'_1\left(\alpha_j - i\ln r_0, q\right)}{\vartheta_1\left(\alpha_j - i\ln r_0, q\right)} \right] = 0.$$

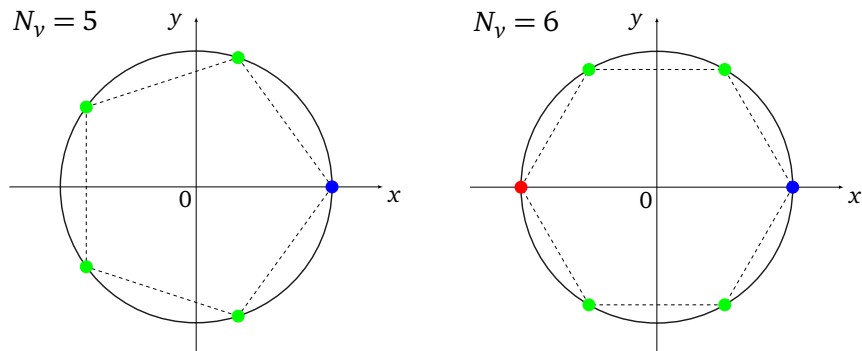

Figure 12: Symmetric necklaces made of odd (left) or even (right) vortices.

To simplify these equations, let us first recall the explicit expression of Jacobi elliptic theta functions and its first derivative

$$\vartheta_1(z,q) = \sum_{n=0}^{\infty} \mathcal{F}_n(q) \sin\left[(2n+1)z\right], \qquad \vartheta_1'(z,q) = \sum_{n=0}^{\infty} (2n+1)\mathcal{F}_n(q) \cos\left[(2n+1)z\right], \quad \text{(D.8)}$$

with the shorthand notation $\mathcal{F}_n(q) = 2(-1)^n q^{(n+1/2)^2}$.

For simplicity we will consider only symmetric configurations like the ones shown in Fig. 12. The black circle represents the circular orbit with radius $r_0$ and the blue dot is the first vortex that we arbitrarily fix at position $(r_1, \theta_1) = (r_0, 0)$.

For an odd number of vortices, the vortices with $j > 1$ [bright green dots in Fig. 12(left)] can be grouped into pairs that are symmetric with respect to the horizontal axis. For each pair, the indices $(j, j')$ satisfy $j + j' = N_\nu + 2$, so that the angular parts in the arguments of theta functions are related by:

$$\alpha_{j'} = \frac{\pi}{N_\nu}(1-j') = -\alpha_j - \pi. \tag{D.9}$$

Two ratios between theta functions appear in Eq. (D.7) and for one of them the two contributions within each pair exactly cancel out, as it easily follows from:

$$\frac{\vartheta_1'(-\alpha_j - \pi, q)}{\vartheta_1(-\alpha_j - \pi, q)} = -\frac{\sum_{n=0}^{\infty}(2n+1)\mathcal{F}_n(q)\cos\left[(2n+1)\alpha_j\right]}{\sum_{n=0}^{\infty}\mathcal{F}_n(q)\sin\left[(2n+1)\alpha_j\right]} = -\frac{\vartheta_1'(\alpha_j, q)}{\vartheta_1(\alpha_j, q)}. \tag{D.10}$$

For the second ratio, instead, one ends up with a purely imaginary term. If we call $\beta_n \equiv \ln\left(r_0^{2n+1}\right)$, in fact, one gets:

$$\begin{aligned}\frac{\vartheta_1'(-\alpha_j - \pi - i\ln r_0, q)}{\vartheta_1(-\alpha_j - \pi - i\ln r_0, q)} &= -\frac{\sum_{n=0}^{\infty}(2n+1)\mathcal{F}_n(q)\cos\left[(2n+1)\alpha_j + i\beta_n\right]}{\sum_{n=0}^{\infty}\mathcal{F}_n(q)\sin\left[(2n+1)\alpha_j + i\beta_n\right]} \\ &= -\left(\frac{\vartheta_1'(\alpha_j - i\ln r_0, q)}{\vartheta_1(\alpha_j - i\ln r_0, q)}\right)^*.\end{aligned} \tag{D.11}$$

When the number of vortices $N_\nu$ is even, there is an additional vortex located at angle $\pi$ [see the red dot in Fig. 12(right)] for which $\alpha_j = -\pi/2$. In this specific point, the two ratios become:

$$\begin{aligned}\frac{\vartheta_1'\left(-\frac{\pi}{2}, q\right)}{\vartheta_1\left(-\frac{\pi}{2}, q\right)} &= -\frac{\sum_{n=0}^{\infty}(2n+1)\mathcal{F}_n(q)\cos\left[(2n+1)\pi/2\right]}{\sum_{n=0}^{\infty}\mathcal{F}_n(q)\sin\left[(2n+1)\pi/2\right]} = 0, \\ \frac{\vartheta_1'\left(-\frac{\pi}{2} - i\ln r_0, q\right)}{\vartheta_1\left(-\frac{\pi}{2} - i\ln r_0, q\right)} &= i\frac{\sum_{n=0}^{\infty}(2n+1)\mathcal{F}_n(q)(-1)^n \sinh\beta_n}{\sum_{n=0}^{\infty}\mathcal{F}_n(q)(-1)^n \cosh\beta_n}.\end{aligned} \tag{D.12}$$

Exactly as for the situation previously discussed, the first contribution vanishes, while the second one gives a purely imaginary term.

It is now clear that the second of Eqs. (D.7) becomes an identity, while the properties of the theta functions allow to reduce the first one to:

$$\frac{\tilde{\mu}}{N_\nu}\Omega_{N_\nu}^2 - 2\Omega_{N_\nu} + \frac{2}{r_0^2}\left[\mathfrak{n}_1 - \frac{1}{2} + \frac{i}{2}\frac{\vartheta_1'(-i\ln r_0, q)}{\vartheta_1(-i\ln r_0, q)} + \frac{i}{2}\sum_{j=2}^{N_\nu}\frac{\vartheta_1'\left(\alpha_j - i\ln r_0, q\right)}{\vartheta_1\left(\alpha_j - i\ln r_0, q\right)}\right] = 0. \quad (\text{D}.13)$$

We denote with $\mathcal{B}(r_0)$ the dimensionless function in the square parenthesis, so that the two roots of the quadratic equation read, in conventional units:

$$\Omega_{N_\nu}^{(\pm)}(r_0) = \frac{\hbar}{m_a R_2^2}\frac{N_\nu}{\tilde{\mu}}\left(1 \pm \sqrt{1 - \frac{2\tilde{\mu}R_2^2}{N_\nu}\frac{\mathcal{B}(r_0)}{r_0^2}}\right). \quad (\text{D}.14)$$

As in the case of an isolated vortex, $\Omega_{N_\nu}^{(-)}$ is the stable solution that in the small mass limit $\mu \to 0$ correctly recovers the result in Eq. (D.3).

Notice that when $\mathfrak{n}_1 = 0$ and in the limit $R_1 \to 0$, using the expansion of $\vartheta_1$ for small $q$, one can verify that Eq. (D.13) reduces to the equation for a necklace of $N_\nu$ vortices inside a circular trap of radius $R_2$, as derived in Ref. [58]:

$$\Omega_{N_\nu}\left(1 - \frac{\tilde{\mu}}{2N_\nu}\Omega_{N_\nu}\right) = \frac{1}{r_0^2}\left(\frac{N_\nu - 1}{2} + N_\nu\frac{r_0^{2N_\nu}}{1 - r_0^{2N_\nu}}\right). \quad (\text{D}.15)$$

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
