# Peer review of "Massive superfluid vortices and vortex necklaces on a planar annulus"

_SciPost Physics, doi:SciPost Phys. 15, 057 (2023)_

## Round 1 · Referee Report · Anonymous (Referee 1) · 2023-3-16

Strengths

This paper models the interesting scenario of the dynamics of superfluid vortices with massive cores that are confined in a toroidal geometry. The primary strength of the work is:
(1) drawing an interesting analogy to the motion of a charged particle in an electromagnetic field, by presenting plasma orbit theory to describe the resulting massive core vortex dynamics when perturbed from their equilibrium state. Other strengths of the work are:
(2) The authors are thorough, comparing their results applying three models, a point vortex formalism, mean field models and in applying plasma orbit theory.
(3) The authors also look at the interesting configuration of a ring of vortices confined in the toroid.
(4) These results should be testable in future superfluid experiments.

Weaknesses

The article has some weaknesses:
( 1) While the results are interesting, the article is at dense to read and it is difficult to follow the derivations and results presented.
(2) It appears that most of the theoretical machinery (massive point vortex models) and some of the results have already been derived in previous work and the novelty of the article lies in drawing the analogy to plasma orbit theory.
(3) The article would benefit from including a discussion of the results using physical intuition (ie interaction between point vortices and image vortices, comparison to point vortex in a channel or around a hard circular boundary) to explain and help the reader understand why the results make sense.
(4) The language used around describing the results and the conditions for dynamics with radial oscillations, 'plasma orbit' and epitrochoid curves could be improved. For example, from the abstract, one could understand that the combination of radial oscillations on top of the usual uniform precession trajectories are equilibrium dynamics of massive vortices. However reading the article it becomes clear that these sorts of trajectories arise when the initial position of the massive vortices are perturbed, and so (please correct me if I am misunderstanding) would better be described as excited state trajectories of massive vortices.

Report

Drawing the comparison to plasma orbit theory to describe the dynamics of vortices with massive cores clearly meets the expectation of SciPost to 'Provide a novel and synergetic link between different research areas' and for this reason I believe this article is suitable for publication following revision. It is also evident that there are clear avenues for future research stemming from the results presented in this article, for example, what do the perturbed dynamics of a necklace of 'pairs' of massive vortices vortices look like? How do the dynamics change when the initial position of only a single vortex in the necklace is perturbed? As well as other avenues for future work discussed by the authors in the article.

Requested changes

I suggest some small changes below:
(1) Considering the points raised earlier, my main suggestion is that the authors try to improve the readability of the article.
(2) Is it appropriate to understand the trajectories that deviate from circular orbits with uniform angular velocity, such as radial oscillation on top of uniform precession as excited state dynamics?
(3) Where it makes sense, include a discussion of dynamics using physical intuition - ie interaction between point vortices and image vortices, comparison to point vortex in a channel or around a hard circular boundary. Relating to this, what is the reason behind the angular velocity in figure 2(a) changing sign with the radius of the circular orbit? Can you also explain the changing sign in angular velocities in later figures? ie 8(a) & 9 (b).
(4)The notation (r,{r_j}) is confusing. Please explain this in the text.
(5) Assumption of a cut-off at the vortex core - does it break at some point with increasing mass?
(6) How would the dynamics change if only the initial position of the massive core was shifted but the vortex was not. Would similar dynamics occur?
(7) How do you define a small displacement and what is the appropriate length-scale to compare to? Vortex core size? The displacement seems quite large in some cases.
(8) In figure 3(a) there is no blue shaded region (which is referred to in the text). I assume you mean the brownish region to the right of the blue line.
(9) On first reading the paper, I found this sentence ambiguous
‘’This regular motion’ becomes unstable beyond a critical vortex mass’.
By ‘this regular motion’ are the authors referring to the radial oscillations around the annulus?
(10)For ease of readability please include the final equation derived in appendices also in the appendix in addition to referring back to the body of the article.

  • validity: good
  • significance: ok
  • originality: good
  • clarity: ok
  • formatting: reasonable
  • grammar: good

Author:  Matteo Caldara  on 2023-04-11  [id 3580]

(in reply to Report 1 on 2023-03-16)

We thank the Referee for her/his careful, constructive and positive report. We also thank her/him for providing further useful comments and suggestions for improving it. We reply point-by-point in the attached pdf document (please see comments to the resubmission).

Attachment:

2023_04_11_Reply_to_Referees_r1hzBHO.pdf

---

## Round 1 · Referee Report · Anonymous (Referee 2) · 2023-3-22

Strengths

(1) Scientifically sound, of a good quality, and addresses interesting open problems within the field.
(2) Approaches the problem at hand with multiple complementary analytical and numerical techniques.
(3) Brings techniques from other fields in order to tackle the problem.
(4) The paper covers a lot of ground: one can understand the importance of mass ratio, system size, perturbation size, number of vortices, etc. from a single work.

Weaknesses

(1) The paper is an extension of Refs. [20,21,30], in a ring rather than a disk, and the appearance of epitrochoidal orbits is expected.
(2) Involving three methods in one paper means that there's a lot of notation to understand and digest.
(3) Some overlapping notation ("a" and "b" are used multiple times in different contexts)

Report

The paper is of high scientific quality, and there is no doubt that this work comprehensively uncovers the physics of the orbits of vortices with an in-filling component, within the potential considered. Looking to the journal's acceptance criteria, of the 6 mandatory requirements there is work required to aid the reader in understanding the work presented (point 1) and more simulation details should be added such that the reader could reproduce the results (point 5). Regarding the expected criteria, the work does provide a novel and synergetic link between different research areas, by linking the observations to plasma orbit theory. Ultimately, I believe the work should be accepted after revisions. Crucially, a paragraph in the introduction needs to properly motivate how this system differs from that covered in Refs. [20,21,30], and what novelty does the inclusion of this inner ring boundary bring. Would the results for the plasma orbit theory worked just as well for the disk condensate case?

Requested changes

(1) How appropriate is the quasi-2D model for the parameters chosen? With the thickness taken, d_z, do you satisfy the condition \mu<< \hbar \omega_z?
(2) The immiscibility condition stated is only exactly true when the atom numbers between the components are equal. Though you are clearly in the immiscible regime, the text should be modified and a citation to the work of K. L. Lee et al., Phys Rev A 94, 013602 (2016) should be added.
(3) I think the equations and model may be a little easier to follow by simply deleting all terms with n_1 (the persistent current). In this work, you always consider n_1=0, and the 91 (!) equations are already tough enough as it is.
(4) In the conclusions it is stated that the GP was extensively tested against the point-vortex model, I think in order for this statement to hold true a comparable GP simulation should also be done to match the onset of vortex expulsion from the condensate.

  • validity: top
  • significance: good
  • originality: ok
  • clarity: ok
  • formatting: perfect
  • grammar: perfect

Author:  Matteo Caldara  on 2023-04-11  [id 3581]

(in reply to Report 2 on 2023-03-22)

We thank the Referee for her/his careful, constructive and positive report. We also thank her/him for providing further useful comments and suggestions for improving it. We reply point-by-point in the attached pdf document (please see comments to the resubmission).

Attachment:

2023_04_11_Reply_to_Referees_Dw7CVFU.pdf

---

## Round 2 · Author Response

thank you very much for sending us the reports of the two Referees. We are grateful to the Referees for their careful, positive and constructive reports that helped us improving our manuscript. We have revised the manuscript complying with the requests of both Referees, and we have replaced the arXiv manuscript with the revised one. Our detailed point-by-point response to the Referees has been uploaded.
In view of the already positive reports of both Referees and of the substantial improvements that we have implemented in response to their comments, we are confident that this revised version of our paper can be accepted for publication on SciPost Physics.
Best regards,
The authors

---

## Round 2 · List of Changes

A detailed list of changes has been attached as a pdf file (please, see Replies to Referees)

---

## Editorial Decision

published